# Early winter barium excess in the Southern Indian Ocean as an annual remineralisation proxy (GEOTRACES GIPr07 cruise)

Natasha René van Horsten[1,2,3], Hélène Planquette[1], Géraldine Sarthou[1], Thomas James Ryan-Keogh[2], Nolwenn Lemaitre[5], Thato Nicholas Mtshali[4], Alakendra Roychoudhury[3], and Eva Bucciarelli[1]

[1]Univ Brest, CNRS, IRD, Ifremer, LEMAR, F-29280 Plouzane, France.
[2]SOCCO, CSIR, Lower Hope Road, Cape Town, South Africa, 7700.
[3]TracEx, Department of Earth Sciences, Stellenbosch University, Stellenbosch, South Africa, 7600.
[4]Department of Forestry, Fisheries and Environment, Oceans and Coast, Foretrust Building, Martin Hammerschlag Way, Cape Town, South Africa, 8001.
[5]Department of Earth Sciences, Institute of Geochemistry and Petrology, ETH Zurich, Zurich, Switzerland.

*Correspondence to*: Natasha van Horsten (natasha.vanhorsten@uct.ac.za), Eva Bucciarelli (Eva.Bucciarelli@univ-brest.fr)

**Abstract.** The Southern Ocean (SO) is of global importance to the carbon cycle, and processes such as mesopelagic remineralisation that impact the efficiency of the biological carbon pump in this region need to be better constrained. During this study early austral winter barium excess ($Ba_{xs}$) concentrations were measured for the first time, along 30°E in the Southern Indian Ocean. Winter $Ba_{xs}$ concentrations of 59 to 684 pmol $L^{-1}$ were comparable to those observed throughout other seasons. The expected decline of the mesopelagic $Ba_{xs}$ signal to background values during winter was not observed, supporting the hypothesis that this remineralisation proxy likely has a longer timescale than previously reported. A compilation of available SO mesopelagic $Ba_{xs}$ data, including data from this study, shows an accumulation rate of ~ 0.9 µmol $m^{-2}$ $d^{-1}$ from September to July that correlates with temporally integrated remotely sensed primary productivity (PP), throughout the SO from data spanning ~ 20 years, advocating for a possible annual timescale of this proxy. The percentage of mesopelagic particulate organic carbon (POC) remineralisation as calculated from estimated POC remineralisation fluxes over integrated remotely sensed PP was ~ 2 fold higher south of the polar front (19 ± 15 %, n = 39) than north of the polar front (10 ± 10 %, n = 29), revealing the higher surface carbon export efficiency further south. By linking integrated remotely sensed PP to mesopelagic $Ba_{xs}$ stock we could obtain better estimates of carbon export and remineralisation signals within the SO on annual and basin scales.

## 1 Introduction

The Southern Ocean (SO) is a carbon sink of global significance responsible for 40 – 50 % of the global oceans' carbon uptake (Friedlingstein et al., 2019; Gregor et al., 2019; Gruber et al., 2019). Oceanic carbon uptake is regulated by various processes, including the biological carbon pump (BCP). Inorganic carbon is consumed and released by photosynthetic organisms through

photosynthesis and respiration (Sarmiento and Gruber, 2006), thereby regulating the earth's carbon cycle by partially sequestering photosynthetically fixed $CO_2$ in the ocean interior (Honjo et al., 2014). In particular, the SO BCP is a crucial contributor to the earth's carbon cycle by exporting, from surface waters, ~ 3 Pg C $yr^{-1}$ of the ~ 10 Pg C $yr^{-1}$ global export production (Schlitzer, 2002). The efficiency of the BCP is linked to the export and preservation of surface particulate matter and is directly linked to atmospheric $CO_2$ levels, on glacial-interglacial timescales (Honjo et al., 2014; Sigman et al., 2010). Sedimentation out of the surface layer (~100 m) is defined as surface export and out of the mesopelagic zone (~1000 m) as deep export (Passow and Carlow, 2012). There are large gaps in our knowledge with regard to deep carbon export, internal cycling and the seasonality of these processes (Takahashi et al., 2012). The magnitude of deep carbon export is dependent on the efficiency of mesopelagic remineralisation (Jacquet et al., 2015) which can balance or even exceed particulate organic carbon (POC) surface export, especially later in the growing season, thereby limiting deep export (Buesseler and Boyd, 2009; Cardinal et al., 2005; Jacquet et al., 2011, 2015; Lemaitre et al., 2018; Planchon et al., 2013). A possible explanation for imbalances between surface export and mesopelagic processes can be lateral advection of surface waters with lower particle export relative to the mesopelagic signal (Planchon et al., 2013). It is also possible that continued remineralisation of earlier larger export fluxes is detected in the mesopelagic signal but not in the export fluxes of in situ observations (Planchon et al., 2013). In addition to this, the efficiency of remineralisation is influenced by the size and composition of exported particles (Rosengard et al., 2015; Twining et al., 2014) as well as the pathway by which these particles are transported downwards (e.g., eddy-subduction, active migration, sinking or mixing) from the surface mixed layer to the mesopelagic zone (Boyd et al., 2019; Le Moigne, 2019), creating an intricate web of processes to disentangle. Mesopelagic remineralisation has also been shown to be influenced by environmental factors, such as temperature, phytoplankton community structure and nutrient availability (Bopp et al., 2013; Buesseler and Boyd, 2009). Indeed, nutrient limitation in surface waters limits export and consequently mesopelagic remineralisation by promoting the shift to smaller phytoplankton assemblages that preferentially take up recycled nutrients in the surface mixed layer (Planchon et al., 2013). Phytoplankton community composition exerts an important control where diatoms are more efficiently exported, due to their large size and ballasting by biogenic silica, compared to smaller non-diatom phytoplankton (Armstrong et al., 2009; Buesseler, 1998; Ducklow et al., 2001). Latitudinal trends in remineralisation efficiency can also be linked to temperature-dependent heterotrophs that are responsible for remineralisation (DeVries and Weber, 2017; Marsay et al., 2015). The mesopelagic layer is under-studied, especially in the high latitudes, and therefore these processes are poorly constrained, despite their importance to global elemental cycles, including that of carbon (Le Moigne, 2019; Robinson et al., 2010).

Export and remineralisation tracers, such as $^{234}$Th/$^{238}$U and apparent oxygen utilisation (AOU), have been used to study mesopelagic POC remineralisation fluxes (Buesseler et al., 2005; Planchon et al., 2013; Lemaitre et al., 2018). Surface export is set by the deficit of $^{234}$Th activities over $^{238}$U activities, while remineralisation processes are reflected by $^{234}$Th/$^{238}$U ratios larger than 1 below the surface mixed layer integrating processes over a 2 to 3 week period (Buesseler et al., 2005; Planchon et al., 2013). AOU is the depletion of oxygen ($O_2$) in the ocean interior relative to surface saturation, due to biological respiration, when surface water masses are subducted. AOU is dependent on salinity and temperature and integrates

remineralisation on timescales of years to decades (Ito et al., 2004). Inaccuracies have, however, been detected with AOU as
a remineralisation proxy, specifically in high latitude areas, due to $O_2$ undersaturation as a consequence of large temperature
gradients (Ito et al., 2004).
Barium excess ($Ba_{xs}$) is another proxy utilised to yield estimates of mesopelagic POC remineralisation fluxes. It is defined as
the "biogenic" portion of particulate barium (pBa) as barite crystals, formed by the decay of bio-aggregates below the surface
mixed layer (Bishop, 1988; Dehairs et al., 1980; Lam and Bishop, 2007; Legeleux and Reyss, 1996; van Beek et al., 2007). As
these crystals are released, a $Ba_{xs}$ peak is formed within the mesopelagic zone which has been found to correlate to primary
production (PP), $O_2$ consumption and POC remineralisation (Dehairs and Goeyens, 1996; Dehairs et al., 1997). Depth-
integrated rates of $O_2$ consumption between the base of the mixed layer and 1000 m were estimated using an inverse 1-D
advection-diffusion-consumption model (Shopova et al., 1995) to develop transfer functions between the $Ba_{xs}$ signal and the
rate of surface POC export for subsequent mesopelagic remineralisation (Dehairs and Goeyens, 1996; Dehairs et al., 1997).
Strong correlations have been obtained between the well-established export/remineralisation flux proxy [234]Th and $Ba_{xs}$, during
studies conducted in the SO and the North Atlantic, confirming the validity of $Ba_{xs}$ as a remineralisation proxy (Cardinal et
al., 2005; Lemaitre et al., 2018; Planchon et al., 2013). Estimates of POC remineralisation fluxes, using the $Ba_{xs}$ proxy, are
directly influenced by the background signal of $Ba_{xs}$, after partial dissolution and sedimentation from the previous bloom
season. It can be thought of as "pre-formed" $Ba_{xs}$, defined as the $Ba_{residual}$ signal at zero $O_2$ consumption (Jacquet et al., 2015).
Because studies conducted in spring and summer suggest that the mesopelagic $Ba_{xs}$ signal lasts between a few days to a few
weeks (Dehairs et al., 1997; Cardinal et al., 2005; Jacquet et al., 2007, 2008a), it is postulated that winter measurements should
give the true SO $Ba_{residual}$ value (Jacquet et al., 2008b, 2011). In this context, as part of a GEOTRACES process study (GIpr07)
of a transect along 30°E in the Southern Indian Ocean (58.5°S to 41.0°S), we studied $Ba_{xs}$ distributions during early austral
winter (July 2017) to better constrain the SO $Ba_{residual}$ concentrations and the timescale of this proxy. To our knowledge these
are the first reported wintertime values for this proxy in the SO.

## 2    Materials and Methods

### 2.1    Sampling and hydrography

During the GEOTRACES GIpr07 cruise, which took place in early austral winter (28 June - 13 July 2017) onboard the R/V
*SA Agulhas II*, seven stations were sampled along 30°E, from 58.5°S to 41.0°S (WOCE I06S, Figure 1a). At each station
between 15 and 21 samples were collected from 25 m down to 1500 m, for shallow stations, and down to 4250 m, for deep
stations, to be analysed for multiple parameters.
Positions of the fronts during the cruise were determined using the July monthly mean absolute dynamic topography data from
the CLS/AVISO product (Rio et al., 2011), with boundary definitions from Swart et al. (2010). From north to south the
identified fronts are, the Subtropical Front (STF), the Subantarctic Front (SAF), the Polar Front (PF), the Southern Antarctic
Circumpolar Current Front (SACCf) and the Southern Boundary (SBdy) (Figure 1a). The marginal ice zone, identified as the
position of 30 % ice cover, was positioned at 61.7°S, approximately 3° (356 km) south of the southernmost station (de Jong et
al., 2018). Therefore, a potential sea ice influence on our study area can be disregarded.

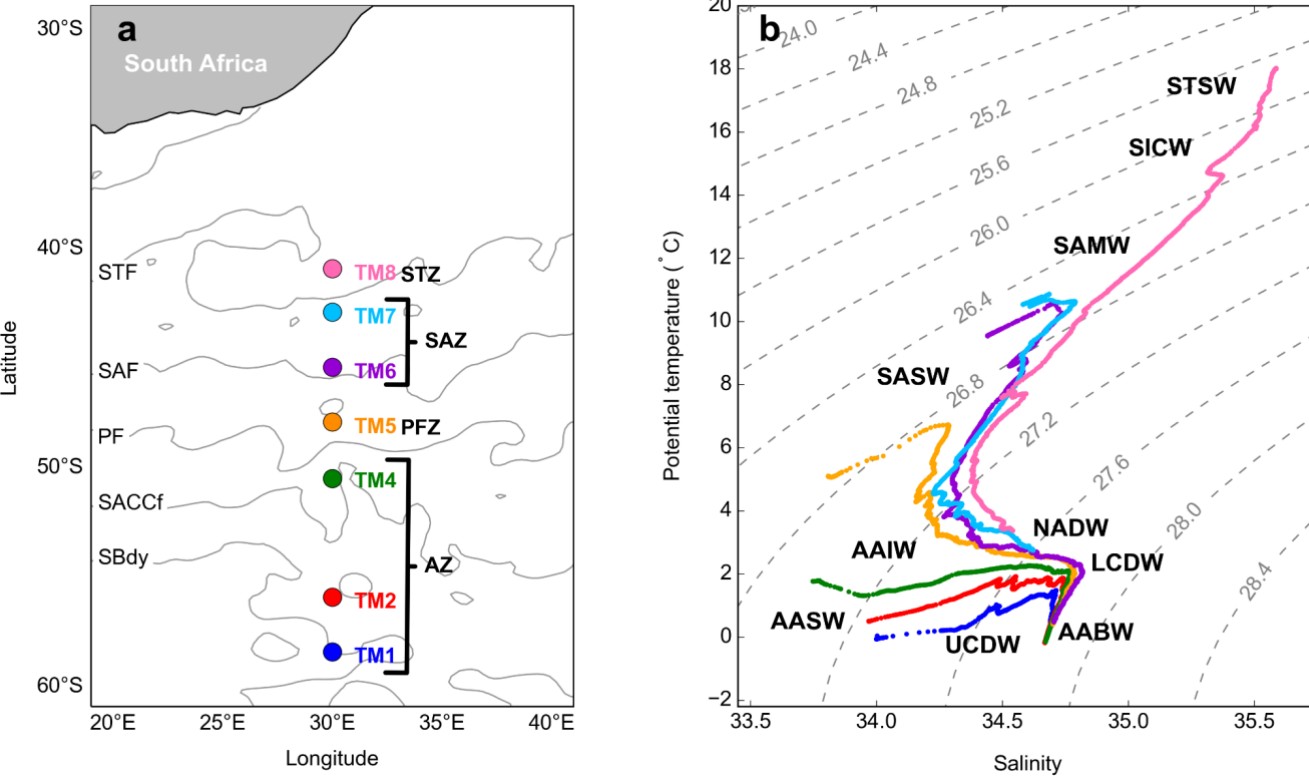

**Figure 1: (a) GEOTRACES GIPr07 cruise sampling stations overlaid on a map with frontal positions; namely, the Subtropical Front**
**(STF), the Subantarctic Front (SAF), the Polar Front (PF), the Southern Antarctic Circumpolar Current Front (SACCf) and the**
**Southern Boundary (SBdy), as determined by mean absolute dynamic topography (MADT) and crossing over four zones; namely,**
**the Antarctic zone (AZ), the Polar frontal zone (PFZ), the Subantarctic zone (SAZ) and the Subtropical zone (STZ). (b) Potential**
**temperature plotted against salinity, overlaid on isopycnals and identification of water masses sampled; namely, Subtropical Surface**
**Water (STSW), South Indian Central Water (SICW), Subantarctic Mode Water (SAMW), Subantarctic Surface Water (SASW),**
**Antarctic Intermediate Water (AAIW), Antarctic Surface Water (AASW), North Atlantic Deep Water (NADW), Lower**
**Circumpolar Deep Water (LCDW), Upper Circumpolar Deep Water (UCDW), and Antarctic Bottom Water (AABW).**
**2.2    Temperature, salinity and dissolved O$_2$**
Temperature (°C), salinity and dissolved O$_2$ (µmol L$^{-1}$) profiles were measured by sensors (SBE 911plus) which were
calibrated by the manufacturer within a year prior to the cruise. At each cast, discrete seawater samples were collected and
analysed onboard for in situ calibration of sensor data for salinity (8410A Portasal salinometer, $R^2 = 0.99$) and dissolved O$_2$
concentrations (Metrohm 848 titrino plus, $R^2 = 0.83$; Ehrhardt et al., 1983). Temperature and salinity measurements were used
to calculate potential density ($\sigma_\theta$; Gill, 1982) to characterise water masses sampled and to identify the mixed layer depth
(MLD). The MLD is the depth at which there is a change of 0.03 kg m$^{-3}$ in $\sigma_\theta$ from a near-surface value at ~ 10 m (de Boyer
Montégut, et al., 2004). Decreases in dissolved O$_2$ concentrations at intermediate depths, together with Ba$_{xs}$ concentrations,
were used to define the mesopelagic remineralisation layer.

## 2.3 pBa and particulate aluminium

Profile sampling of the water column was conducted with a GEOTRACES compliant trace metal clean CTD housed on an epoxy coated aluminium frame with titanium bolts equipped with 24 x 12 L trace metal clean Teflon coated GO-FLO bottles (General Oceanics). All sampling and analyses were conducted following the GEOTRACES clean sampling and analysis protocols (Cutter et al., 2017). Volumes of 2 to 7 L seawater were filtered from the GO-FLO bottles onto acid-washed polyethersulfone filters (25 mm diameter, Supor, 0.45 µm pore size), for pBa and particulate aluminium (pAl) analyses. Filters were mounted in-line on the side spigot of each Go-Flo bottle, on swinnex filter holders. Furthermore, bottles were mixed 3 times before filtration, as recommended by the GEOTRACES protocols (Cutter et al., 2017), to ensure homogenous sampling. Although the large fast-sinking fraction of particles may be under-sampled by using bottles (Bishop and Edmond, 1976; Planquette and Sherrell, 2012), comparing data that were generated using the same, internationally validated sampling systems and protocols (Cutter et al. 2017), as we do in this study, minimises potential bias. After filtration, filters were placed in trace metal clean petri slides (Pall) and kept frozen at -20°C until further processing on land. Sample processing was conducted under a class 100 HEPA filtered laminar flow and extraction hood in a clean laboratory.

The pBa and pAl samples were processed and analysed 6 months after sample collection, at LEMAR (France). Unused blank filters and filters containing the samples were acid reflux digested at 130°C in acid-cleaned savillex vials using a mixture of HF and $HNO_3$ (both Ultrapure grade, Merck) solutions (Planquette and Sherrell, 2012). Archive solutions were stored in 3 ml of 0.12 M $HNO_3$ (Ultrapur grade), of which 250 µL was diluted up to 2 mL for analysis by sector field inductively coupled plasma mass spectrometry (SF-ICP-MS, Element XR Thermo Scientific). Samples were spiked with 1 µg $L^{-1}$ indium as an internal standard to correct for instrument drift. The detection limits, defined as three times the standard deviation of the blanks (unused filter blanks), were 0.39 pmol $L^{-1}$ and 0.03 nmol $L^{-1}$ (n = 5) for pBa and pAl, respectively. Mean amounts (in nmol) of a given element determined in unused filter blanks were subtracted from the amounts in the sample filter then divided by the volume filtered. Three certified reference materials (BCR 414, MESS 4 and PACS 3) were processed and analysed with the samples to assess the accuracy of the methodology. Our values were in good agreement with the certified values of the reference materials (Table 1) (Jochum et al., 2005). Percentage error of analyses was determined by the repeat analysis of random samples during each run, the mean percentage error of sample analysis for pBa and pAl was 9.2 ± 2.5 % and 11.1 ± 4.6 % (mean ± SD, n = 6), respectively.

**Table 1: Certified reference material recovery data for accuracy determination of pBa and pAl analyses**
**N/A refers to instances where there are no certified values available to check for accuracy**

|  | pBa (mg/kg) | pAl (mg/kg) |
|---|---|---|
| PACS 3 certified (mean ± SD) | N/A | 65800 ± 1700 |
| PACS 3 measured (mean ± SD) | N/A | 73156 ± 15416 |
| PACS 3 mean % recovery | N/A | 111 ± 23 |

| | | |
|---|---|---|
| MESS 4 certified | 920 | $79000 \pm 2000$ |
| MESS 4 (mean $\pm$ SD) | $1033 \pm 28$ | $100048 \pm 26870$ |
| MESS 4 mean % recovery $\pm$ SD | $112 \pm 3$ | $127 \pm 34$ |
| BCR 414 indicative values | $32 \pm 5$ | $2384 \pm 652$ |
| BCR 414 (mean $\pm$ SD) | $34 \pm 4$ | $2651 \pm 317$ |
| BCR 414 mean % recovery $\pm$ SD | $105 \pm 12$ | $111 \pm 13$ |


## 2.4 $Ba_{xs}$ as a proxy for mesopelagic POC remineralisation

The non-lithogenic fraction of pBa, $Ba_{xs}$, was calculated by subtracting the lithogenic fraction of pBa from the total pBa
measured using Eq. 1. The lithogenic contribution to pBa was calculated by multiplying the pAl concentration with the Ba/Al
upper continental crust (UCC) ratio, 0.00135, as determined by Taylor and McLennan (1985).

$$Ba_{xs} = [pBa] - ([pAl] \times (Ba/Al)_{UCC}) \tag{1}$$

Total pBa and $Ba_{xs}$ profiles were nearly identical with a mean percentage $Ba_{xs}$ to total pBa of $99 \pm 1$ % (mean $\pm$ SD, n = 124;
Table S1), indicating that pBa from lithogenic sources was negligible. This ensures the accurate estimation of $Ba_{xs}$, which
requires that less than 50 % of pBa should be associated with lithogenic inputs (Dymond et al., 1992).
The mesopelagic POC remineralisation flux was estimated using Eq. 2 (Dehairs and Goeyens, 1996; Shopova et al., 1995):

$$Mesopelagic\ POC\ remineralisation = Z \times JO_2 \times (C{:}O_2)_{Redfield\ Ratio} \times 12.01 \tag{2}$$

Where the mesopelagic POC remineralisation flux is expressed in mg C $m^{-2}$ $d^{-1}$, Z is the depth range of the mesopelagic $Ba_{xs}$
layer (100 - 1000 m), $C{:}O_2$ is the stoichiometric molar ratio of carbon to $O_2$ consumption by remineralisation as per the Redfield
Ratio (127/175, Broecker et al., 1985), 12.01 is the molar mass of carbon (g $mol^{-1}$) and $JO_2$ is the rate of $O_2$ consumption (µmol
$L^{-1}$ $d^{-1}$) as estimated using Eq. 3:

$$JO_2 = (Mesopelagic\ Ba_{xs} - Ba_{residual})/17200 \tag{3}$$

Eq. 3 (Dehairs and Goeyens, 1996; Shopova et al., 1995) is the linearisation of the exponential function by Dehairs et al.
(1997). Mesopelagic $Ba_{xs}$ is the depth-weighted average $Ba_{xs}$ of the mesopelagic zone (pmol $L^{-1}$), the constant value of 17200
is the slope of the linear regression of depth-weighted average $Ba_{xs}$ (pmol $L^{-1}$) versus $O_2$ consumption rate (µmol $L^{-1}$ $d^{-1}$) and
$Ba_{residual}$ is the deep ocean background value of $Ba_{xs}$ at zero oxygen consumption. The literature value of 180 pmol L$^{-1}$ was
used as the $Ba_{residual}$ value (Jacquet et al, 2008a; 2008b; 2011; 2015; Planchon et al., 2013) in our calculations.
The integrated mesopelagic $Ba_{xs}$ stock ($\mu$mol m$^{-2}$) over the mesopelagic layer (100 - 1000 m) was calculated from the depth-
weighted average $Ba_{xs}$ in order to investigate the link between the accumulated mesopelagic signal and the corresponding
integrated remotely sensed primary productivity (PP).

## 2.5 Integrated remotely sensed PP

The integrated remotely sensed PP (mg C m$^{-2}$ d$^{-1}$) within the surface mixed layer was calculated using the CbPM algorithm
(Behrenfeld et al., 2005), which requires chlorophyll concentration (mg m$^{-3}$), particulate backscatter ($\lambda$ 443 nm, m$^{-1}$),
photosynthetically active radiation (PAR; $\mu$mol photons m$^{-2}$ d$^{-1}$) and the MLD (m). Ocean Colour-Climate Change Initiative
(OC-CCI) data (https://esa-oceancolour-cci.org/), which blends existing data streams into a coherent record, meeting the
quality requirements for climate assessment (Sathyendranath et al., 2019), were used for chlorophyll and particulate
backscatter. PAR was taken from GLOB colour (http://www.globcolour.info/), and the MLD was taken from the climatology
of de Boyer Montegut et al. (2004). The integrated remotely sensed PP data were regridded to 0.25° spatially, using bilinear
interpolation using the Python programming package xESMF (Zhuang, 2018), and averaged monthly. The area-averaged PP
was averaged over a 6 x 1° rectangular sample area, positioned 6° upstream longitudinally, and 1° latitudinally centred around
each sampled station (see discussion for details). In order to assess the validity of the remotely sensed PP data and demonstrate
no meridional bias across the SO, the percentage valid pixels was calculated for data north ($90 \pm 20$ %; mean $\pm$ SD, n = 370)
and south ($82 \pm 29$ % mean $\pm$ SD, n = 488) of the PF, revealing no bias.

## 2.6 Integrated % POC remineralised

The integrated remineralised POC (mg C m$^{-2}$) was estimated by multiplying the POC remineralisation flux (mg C m$^{-2}$ d$^{-1}$), as
estimated using the $Ba_{xs}$ proxy method, by the number of days over which the corresponding remotely sensed PP (mg C m$^{-2}$ d$^{-1}$
$^{1}$) was subsampled. The % POC remineralised was then estimated as the percentage of integrated remotely sensed PP (mg C
m$^{-2}$) remineralised, assuming that the mesopelagic $Ba_{xs}$ stock signal observed is due to the remineralisation of the integrated
surface PP signal.

## 2.7 Statistical analysis

For statistical analysis, the least squares regression method was applied for assessment of significant correlations (Barbur et
al., 1994). Significant differences between regions and regressions were tested using Welch's t-test, with an alpha of 0.05 (95
% confidence level) (Kokoska and Zwillinger, 2000).

## 3    Results

### 3.1    Hydrography

The potential temperature ($\theta$) and salinity (S) along the transect ranged from -0.06 to 18.03 °C and from 33.77 to 35.59, respectively. Where surface $\theta$ and S define four hydrographic zones; namely, the Antarctic zone (AZ; $\theta < 2.5$ °C; $S \leq 34$) from ~ 50°S to 58.5°S, the polar frontal zone (PFZ; $\theta \cong 5$ °C; $S \cong 33.8$) at ~ 48°S, the subantarctic zone (SAZ; $5 < \theta < 11$ °C; $33.8 < S < 34.7$) between 43°S and 45.5°S, and the subtropical zone (STZ; $\theta \geq 17.9$ °C; $S \cong 35.6$) at 41°S (Figure 1a; Anilkumar and Sabu, 2017; Orsi et al., 1995; Pollard et al., 2002). The MLDs along the transect ranged between 97 and 215 m ($144 \pm 39$ m; mean $\pm$ SD, n = 7), shoaling towards the PF (Table S2).

As can be observed on the T-S plot of stations sampled (Figure 1b), different water masses were sampled along the transect throughout the water column. South of the polar front (SPF; $\gtrapprox 50$°S; TM1, 2 and 4), from surface to depth, Antarctic Surface Water (AASW; $27 < \sigma_\theta < 27.4$ kg.m$^{-3}$), Upper and Lower Circumpolar Deep Water (UCDW; $27.2 < \sigma_\theta < 27.75$ kg.m$^{-3}$ and LCDW; $27.75 < \sigma_\theta < 27.85$ kg.m$^{-3}$, respectively), and Antarctic Bottom Water (AABW; $27.8 < \sigma_\theta < 27.85$ kg.m$^{-3}$) were characterised. North of the polar front (NPF) and south of the STF ($< 50$°S; TM5, 6 and 7), from surface to depth, Subantarctic Surface Water (SASW; $26.5 < \sigma_\theta < 26.75$ kg.m$^{-3}$), Antarctic Intermediate Water (AAIW; $26.7 < \sigma_\theta < 27.4$ kg.m$^{-3}$), North Atlantic Deep Water (NADW; $27 < \sigma_\theta < 27.85$ kg.m$^{-3}$) and, as far north as 45.5°S, AABW close to the ocean floor, were identified. At the northernmost station (TM8; 41°S), in the STZ, the water masses sampled include Subtropical Surface Water (STSW; $\sigma_\theta \cong 25.7$ kg.m$^{-3}$), South Indian Central Water (SICW; $25.8 < \sigma_\theta < 26.2$ kg.m$^{-3}$), Subantarctic Mode Water (SAMW; $26.2 < \sigma_\theta < 26.6$ kg.m$^{-3}$), AAIW and NADW.

### 3.2    Dissolved O$_2$

The water column dissolved O$_2$ concentrations ranged from 159 to 333 µmol L$^{-1}$ (Figure 2). Maximum concentrations were observed in the surface mixed layer, increasing southwards along the transect, with a mean value of $287 \pm 40$ µmol L$^{-1}$ (mean $\pm$ SD, n = 700). A decrease in concentrations below the MLD coincided with an increase in $\sigma_\theta$. South of the PF, the decrease in dissolved O$_2$ concentrations at the MLD was sharp and relatively shallow when compared to profiles NPF, which were more gradual, spanning a wider depth range. Within the mesopelagic zone concentrations decreased down to $204 \pm 29$ µmol L$^{-1}$ (mean $\pm$ SD, n = 6373), then remained relatively uniform below 1000 m at $192 \pm 113$ µmol L$^{-1}$ (mean $\pm$ SD, n = 12950).

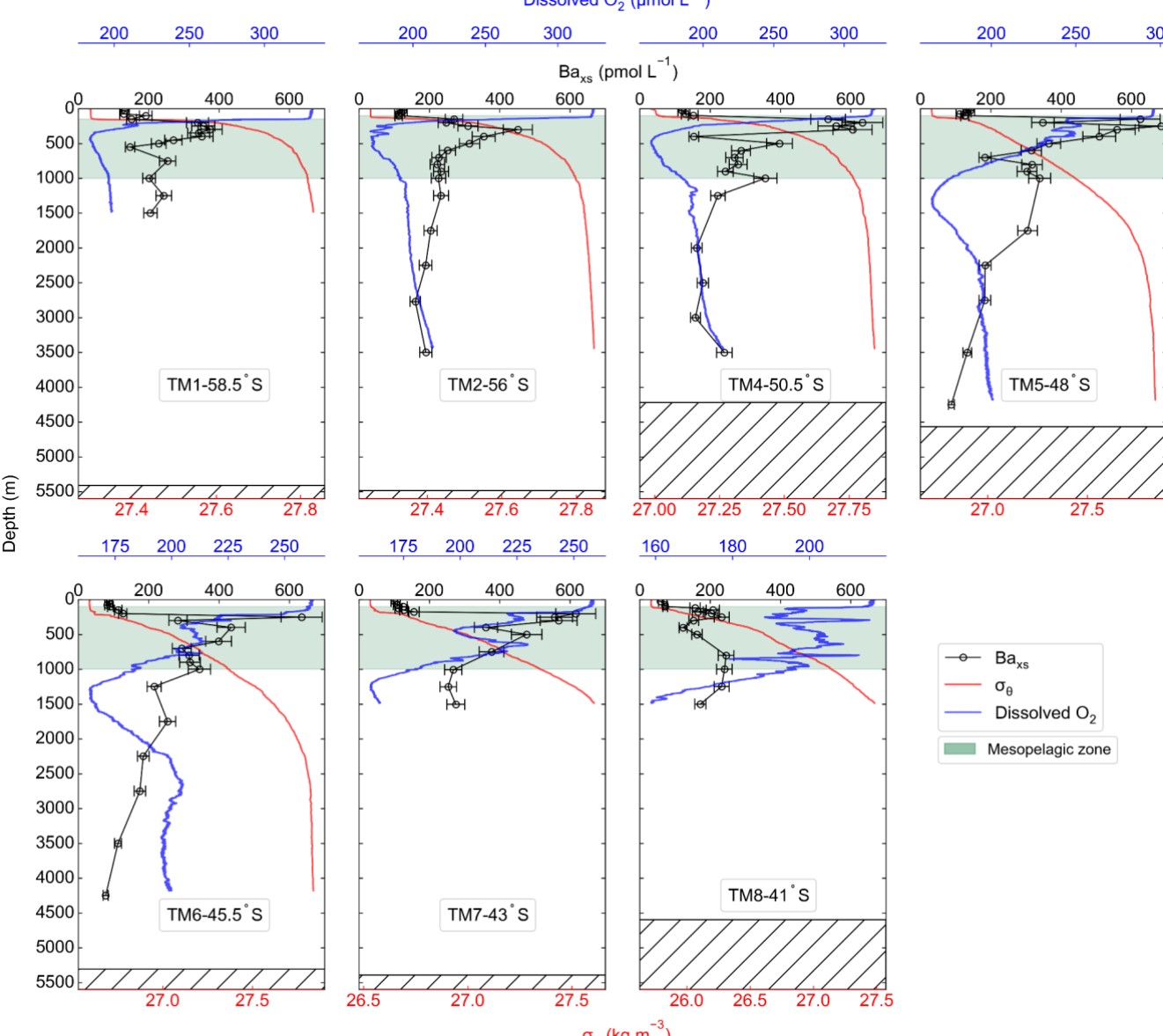

**Figure 2: Ba$_{xs}$ (black circles) with error bars, potential density ($\sigma_\theta$; red) and dissolved O$_2$ (blue) profiles sampled along the transect,**
**plotted against depth, for stations TM1 to TM8, from south to north. The green shaded area is the mesopelagic zone, and the hatched**
**area is the ocean floor.**
### 3.3    Ba$_{xs}$ and estimated POC remineralisation fluxes
Along the transect, Ba$_{xs}$ concentrations ranged from 59 to 684 pmol L$^{-1}$. All profiles exhibited a depletion of Ba$_{xs}$ in the upper
surface waters (59 - 152 pmol L$^{-1}$), then a rapid increase below the MLD (~ 150 m), with concentrations ranging between 113
and 684 pmol L$^{-1}$ in the mesopelagic zone (100 - 1000 m, Figure 2). At the two southernmost stations (TM1 and TM2),
mesopelagic $Ba_{xs}$ peaks spanned a narrower depth range (100 - 600 m) than stations further north, with concentrations reaching
values of ~ 400 pmol $L^{-1}$. Concentrations were higher in the PFZ and SAZ with a maximum of 684 pmol $L^{-1}$ in the PFZ, at
48°S (TM5). The subsurface increase of $Ba_{xs}$ started at slightly deeper depths (150 - 200 m) and spanned wider depth ranges
down to 1000 m, at stations north of the PF. The STZ station, at 41°S (TM8), had the lowest concentrations, only increasing
up to ~ 200 pmol $L^{-1}$. Double peaks were observed at all stations north of the PF, with a shallow and more substantial peak
occurring in the upper mesopelagic zone and a second peak in the lower mesopelagic zone. Below the mesopelagic zone, $Ba_{xs}$
concentrations decreased down to ~ 180 pmol $L^{-1}$ and remained relatively uniform.
The mean $Ba_{residual}$ concentration south of the PF was $183 \pm 29$ pmol $L^{-1}$ (mean ± SD, n = 7), whereas it was $142 \pm 45$ pmol $L^{-1}$
(mean ± SD, n = 8) between the PF and the STF. The two regions were however not significantly different to each other
when conducting a Welch's t-test (t-statistic = 2.10; p-value = 0.06) and when averaging all concentrations below 2000 m
along the transect, the $Ba_{residual}$ concentration was $161 \pm 43$ pmol $L^{-1}$ (mean ± SD, n = 15). This concentration is not statistically
different from the literature value of 180 pmol $L^{-1}$ (Jacquet et al, 2008a; 2008b; 2011; 2015; Planchon et al., 2013), which is
widely used for estimates of POC remineralisation fluxes. For a better comparison with these previous estimates, we used 180
pmol $L^{-1}$ for the $Ba_{residual}$ concentration in our calculations.
The estimated POC remineralisation fluxes for the study area ranged from 6 to 96 mg C $m^{-2}$ $d^{-1}$ (Table S3), increasing
northwards from the southernmost station up to the PFZ from 32 to 92 mg C $m^{-2}$ $d^{-1}$, then decreasing down to 70 mg C $m^{-2}$ $d^{-1}$
at the SAF. The highest flux was estimated in the SAZ, and the lowest flux was estimated in the STZ.
**4 Discussion**
**4.1 Early wintertime $Ba_{xs}$ and $Ba_{residual}$ concentrations**
A noticeable difference between profiles sampled early in the bloom season (Dehairs et al., 1997; Jacquet et al., 2015) versus
those sampled later (Cardinal et al., 2001; Planchon et al., 2013) is the contrasted $Ba_{xs}$ concentrations in the surface mixed
layer. Dehairs et al. (1997) has shown that these concentrations of $Ba_{xs}$ can be as high as 9000 pmol $L^{-1}$ in areas of high
productivity during spring, which then become depleted to concentrations below the SO $Ba_{residual}$ value of ~ 180 pmol $L^{-1}$, as
productivity declines and surface POC export increases (Planchon et al., 2013). These high surface concentrations are,
however, not due to the same process as the one that controls the $Ba_{xs}$ concentrations within the mesopelagic zone (Jacquet et
al., 2011). Surface water concentrations are associated with Ba adsorbed onto particles whereas the mesopelagic $Ba_{xs}$ signal is
due to barite crystals formed within decaying bio-aggregates (Cardinal et al., 2005; Lam and Bishop, 2007; Lemaitre et al.,
2018; Sternberg et al., 2005). In this study, we observed surface depletion of $Ba_{xs}$ at all stations, in line with the assumption
that the bulk surface export from the preceding bloom had been achieved at the time of sampling and, the majority of the $Ba_{xs}$
had been transferred to the mesopelagic zone.
A sharp increase in $\sigma_\theta$ observed at the MLD has previously been identified as the depth at which decaying bio-aggregates are
formed (Lam and Bishop, 2007). These increases coincided with an increase in $Ba_{xs}$ (Figure 2), linking the subsurface $Ba_{xs}$

signal to decaying bio-aggregates as per previous studies (Cardinal et al., 2005; Dehairs et al., 1997; Jacquet et al., 2011). Additionally, decreases observed in dissolved $O_2$ profiles along the transect were also accompanied by coinciding increases in $Ba_{xs}$, in line with $O_2$ consumption due to remineralisation within the mesopelagic zone (Figure 2) (Cardinal et al., 2005; Jacquet et al., 2005, 2011). The observed range of mesopelagic $Ba_{xs}$ concentrations (113 - 684 pmol $L^{-1}$) were comparable to those previously reported in SO open waters (~ 200 - 1000 pmol $L^{-1}$; Cardinal et al., 2001, 2005; Jacquet et al., 2005, 2008a, 2008b, 2011, 2015; Planchon et al., 2013).

$Ba_{xs}$ profiles exhibited similar distributions to those reported throughout bloom seasons in the SO, with distinct peaks observed within the mesopelagic zone at all stations. Earlier in the bloom season, peaks mostly occur within the upper half of the mesopelagic zone (100 - 500 m: Cardinal et al., 2001, 2005; Jacquet et al., 2005, 2008a, 2011, 2015), but as the season progresses, they deepen down towards the bottom half of the mesopelagic zone (500 - >1000 m: Jacquet et al., 2008b, Planchon et al., 2013). Deepening and widening of the remineralisation depth range can be expected as the season progresses, due to continued remineralisation taking place as particles sink to the bottom of the mesopelagic zone (Lemaitre et al., 2018; Planchon et al., 2013). This is also what we observed during early winter at stations NPF, with a second peak in deeper waters, as observed by Jacquet et al. (2008b) during the iron (Fe) fertilisation experiment (EIFEX). The deeper peak could also be linked to relatively larger cells that sink faster as they remineralise, possibly a large bloom earlier in the season.

A distinct latitudinal trend in mesopelagic depth-weighted average $Ba_{xs}$ has generally been observed in the SO with the highest values in the PFZ, decreasing north and southwards from the PF. These latitudinal trends tend to be accompanied by a coinciding trend in in situ surface biomass measurements (Cardinal et al., 2005; Dehairs et al., 1997, Jacquet et al., 2011; Planchon et al., 2013). During our early winter study, we observed a similar latitudinal trend in mesopelagic $Ba_{xs}$ stock ($\mu$mol $m^{-2}$), with an increase from the southernmost station up to the PF, then varying around a maximum in the SAZ, down to the lowest value in the STZ, whereas temporally integrated remotely sensed PP increased progressively northwards to a maximum in the STZ (Figure S1). Time of sampling and extended blooms, which are characteristic of the SAZ (Thomalla et al., 2011), could be contributing factors to the higher values observed in PP and mesopelagic $Ba_{xs}$ distributions at stations north of the PF (Figure S1). Contrary to what was expected, the profiles observed during our early winter study still show a significant mesopelagic remineralisation signal, well after the summer bloom termination, which occurred between April and May (Figure 3), as defined by the point in time when community losses outweigh the growth rate (Thomalla et al., 2011).

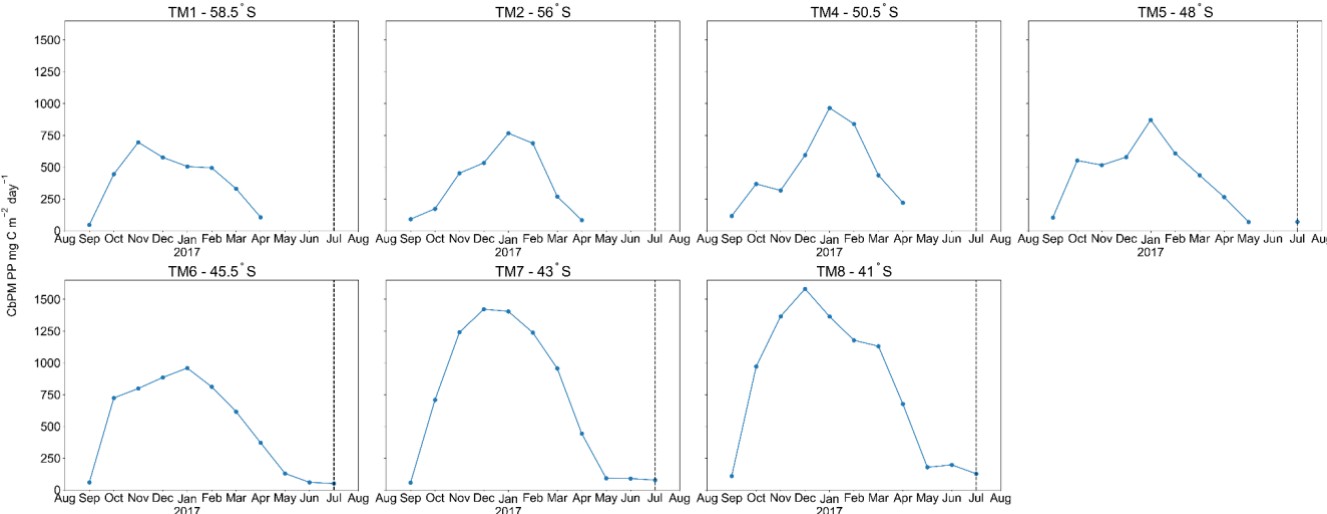

**Figure 3: Time series, area-averaged (6 x 1° rectangular sample area, positioned 6° upstream longitudinally, and 1° latitudinally**
**centred around each station) remotely sensed CbPM-PP (mg C m$^{-2}$ day$^{-1}$), monthly average from 08/2016 to 08/2017, dashed vertical**
**lines indicate sampling date.**
In deeper waters (below 2000 m) along the transect, south of the STF, where remineralisation is minimal compared to the
mesopelagic zone, our Ba$_{xs}$ concentrations of 161 ± 43 pmol L$^{-1}$ (mean ± SD, n = 15) is not significantly different from the
widely used Ba$_{residual}$ concentration of 180 pmol L$^{-1}$, measured during early Spring to late Summer (e.g., Jacquet et al., 2008a;
2008b; 2011; 2015; Planchon et al., 2013). We thus did not observe a wintertime decline to an expected "true" SO background
value, when PP and bacterial activity are suspected to be minimal (Jacquet et al., 2011). There are two possible explanations
for this; firstly, the decline to a winter background signal might never be achieved due to ongoing barite precipitation and
remineralisation, as well as the release of labile Ba attached to phytoplankton as they decay, precipitating into barite crystals,
which could possibly continue throughout winter (Cardinal et al., 2005). Secondly, the low sinking speed of suspended barite
(~ 0.3 m d$^{-1}$, Sternberg et al., 2008), once produced in the mesopelagic layer, implies that it would take ~ 6 years (not
considering reaggregation and redissolution) to sink from 300 m (~ peak of production) to the bottom of the mesopelagic layer
(1000 m). The "true" background value may thus have to be measured at the very end of winter just before the initiation of the
spring bloom. This also suggests that the Ba$_{xs}$ signal in the mesopelagic layer may represent remineralisation activity over
more than a few days to weeks, per previous reports (e.g., Dehairs et al., 1997; Jacquet et al., 2015; Planchon et al., 2013).
**4.2    Timescale of the mesopelagic Ba$_{xs}$ signal**
The Ba$_{xs}$ signal that we observed in winter is in agreement with the suggestion by Dehairs et al. (1997), that there can be
significant carry over between bloom seasons. Other studies have also pointed out that the timescale of this proxy is longer
than a snapshot view (Cardinal et al., 2005) and have highlighted a seasonal increase in mesopelagic Ba$_{xs}$ (Jacquet et al., 2011).
This strongly suggests that the Ba$_{xs}$ signal is not directly linked to synoptic measurements of PP at the time of sampling. In
order to investigate this hypothesis, for the first time, we compiled a SO mesopelagic Ba$_{xs}$ stock dataset with all available

literature data including data from this study (Figure 4a, Table S3). The mesopelagic $Ba_{xs}$ stock was integrated over the $Ba_{xs}$ peak depth range (as identified in each study). As can be seen on the map of the compilation dataset (Figure 4a), these data points were collected across the three basins of the SO, over ~ 20 years. Despite this diversity in observations, a statistically significant accumulation of mesopelagic $Ba_{xs}$ with time is still observed, SPF (Figures 4b) and NPF (Figures 4c). Mesopelagic $Ba_{xs}$ accumulates at a rate of 0.86 ($\pm$0.15) µmol m$^{-2}$ d$^{-1}$ SPF ($R^2 = 0.43$, p-value < 0.05, n = 43; Figure 4b), and at 0.88 ($\pm$0.20) µmol m$^{-2}$ d$^{-1}$ NPF ($R^2 = 0.41$, p-value < 0.05, n = 31; Figure 4c), with no statistically significant difference between the two regions (Welch's t-test = 0.24; p-value = 0.80).

A possible link between the integrated mesopelagic $Ba_{xs}$ stock and the corresponding integrated remotely sensed PP was assessed for all studies conducted after September 1997, when remotely sensed PP data became available. To do so, we first estimated that sub millimetre sized aggregates would take ~ 20 days to sink down to 1000 m (considered as the bottom of the mesopelagic zone in this study), using a sinking speed of 50 m d$^{-1}$, that corresponds to an average literature value (50 - 100 m d$^{-1}$: Riebesell et al., 1991; 50 - 430 m d$^{-1}$ around South Georgia: Cavan et al. 2015; mean of ~ 100 m d$^{-1}$ in the SO as reviewed in Laurenceau-Cornec et al., 2015; Marguerite Bay: 10 - 150 m d$^{-1}$: McDonnell and Buesseler, 2010). Assuming a maximum surface current speed of 0.2 m s$^{-1}$ (Ferrari and Nikurashin, 2010), it was estimated that these aggregates would have originated, 346 km west from the station that was sampled for mesopelagic $Ba_{xs}$, ~ 20 days prior. Using this distance, the dimensions of the sample area were set with the southernmost station (TM1) of this study, where degrees of longitude cover the smallest area. For the sake of consistency this sample area was applied to all sampling locations of the considered dataset. The integrated remotely sensed PP (see section 2.5) was then averaged spatially, positioned 6° upstream longitudinally, and 1° latitudinally centred around each station, in order to capture the surface PP that is assumed to translate to the mesopelagic remineralisation and measured $Ba_{xs}$ stock.

The monthly averaged remotely sensed PP, at the time of sampling, was compiled for the considered dataset, and we found that the PP over the growing season (Figure 4d & e) reaches highest values between January and February (day 125 to 175 of the year), thereafter, steadily decreasing to minimal values in July (~ day 310 of the year, i.e., during our study). The mesopelagic $Ba_{xs}$ accumulation over time can, therefore, not be matched with the remotely sensed PP measured during the month of sampling. A possible relationship between mesopelagic $Ba_{xs}$ stock and temporally integrated remotely sensed PP was further investigated by considering longer timescales. Remotely sensed PP of the preceding bloom was temporally integrated from the preceding September, prior to sampling, as the start of the bloom, in general agreement with previous bloom phenology studies (Thomalla et al., 2011), up to one month prior to the sampling date of the study, taking into consideration time needed for export, aggregate formation and barite crystal release through remineralisation (~ 1 month).

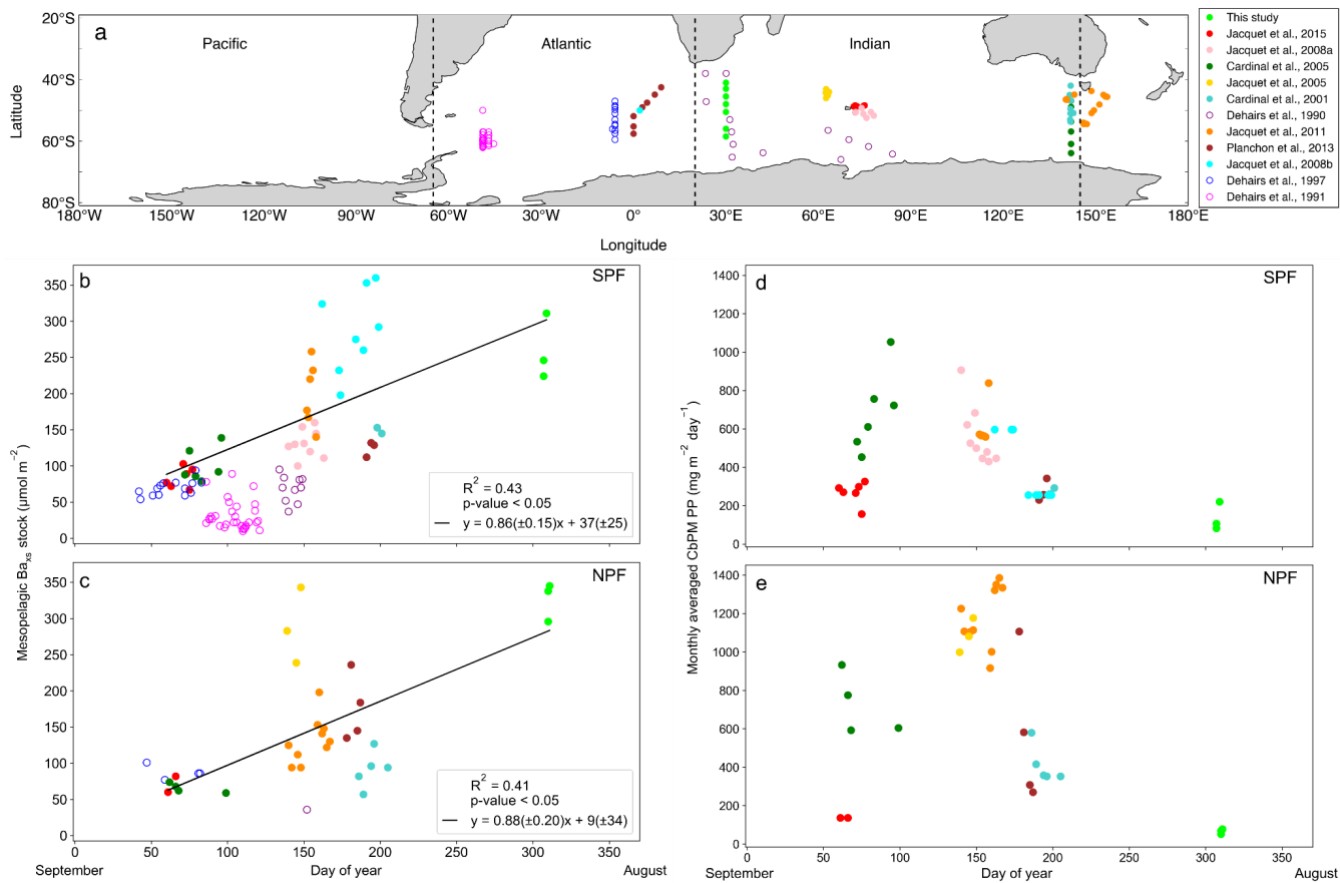

340

**Figure 4: (a) Positions of Ba$_{xs}$ observations compiled from all known SO studies, on a cylindrical equal-area projection of the SO,**
**the three SO basin cut offs are indicated by the dashed black lines, from left to right, Pacific, Atlantic and Indian. Integrated**
**mesopelagic Ba$_{xs}$ stock plotted against day of year sampled, with the 1st of September set as day 1, for all available literature data**
**and winter data from this study. Data was split into two zones using the Polar Front (PF) to divide the SO; (b) South of the PF**
**(SPF) and (c) North of the PF (NPF). Monthly averaged remotely sensed PP plotted against day of year, for locations and dates of**
**the SO compilation dataset and winter data from this study; (d) SPF and (e) NPF. Open circles are data points from studies which**
**did not use HF in the particulate sample digestion procedure, regressions did not include these data, there was, however, no**
**significant difference when including these data points (Table S3).**

Varying timescales were considered between the preceding September up to 1 month prior to sampling (Sept - T1; Table S4),
in monthly increments, that could influence the relationship between remotely sensed PP and the mesopelagic Ba$_{xs}$ stock (Table
S4). The strongest and most significant correlation between the mesopelagic Ba$_{xs}$ stock and integrated remotely sensed PP, for
both north and south of the PF, was obtained from the preceding September up to 1 month prior to sampling (Table S4, Sept -
T1, SPF: Figure 5a, $R^2 = 0.55$, p-value $< 0.05$, n $= 39$; NPF: Figure 5b, $R^2 = 0.42$, p-value $< 0.05$, n $= 31$). When remote sensing
data was limited due to cloud cover and low sunlight during winter months, specifically at the southernmost stations, all
available data was used for the duration of the season. The correlation observed in the STZ is not significant at a 95 %
confidence level (p-value $= 0.10$); however, the limited number of data points (n $= 6$) may preclude any significance from
emerging. The significant positive correlations obtained south of the STF suggest that mesopelagic $Ba_{xs}$ stock can be used as
a remineralisation proxy on an annual timescale instead of only a few weeks. Figure 5 also reveals that for a given PP the
mesopelagic $Ba_{xs}$ stock was 2-fold higher SPF compared to NPF (Welch's t-test, t-statistic = 2.24; p-value < 0.05), this is
further discussed below.

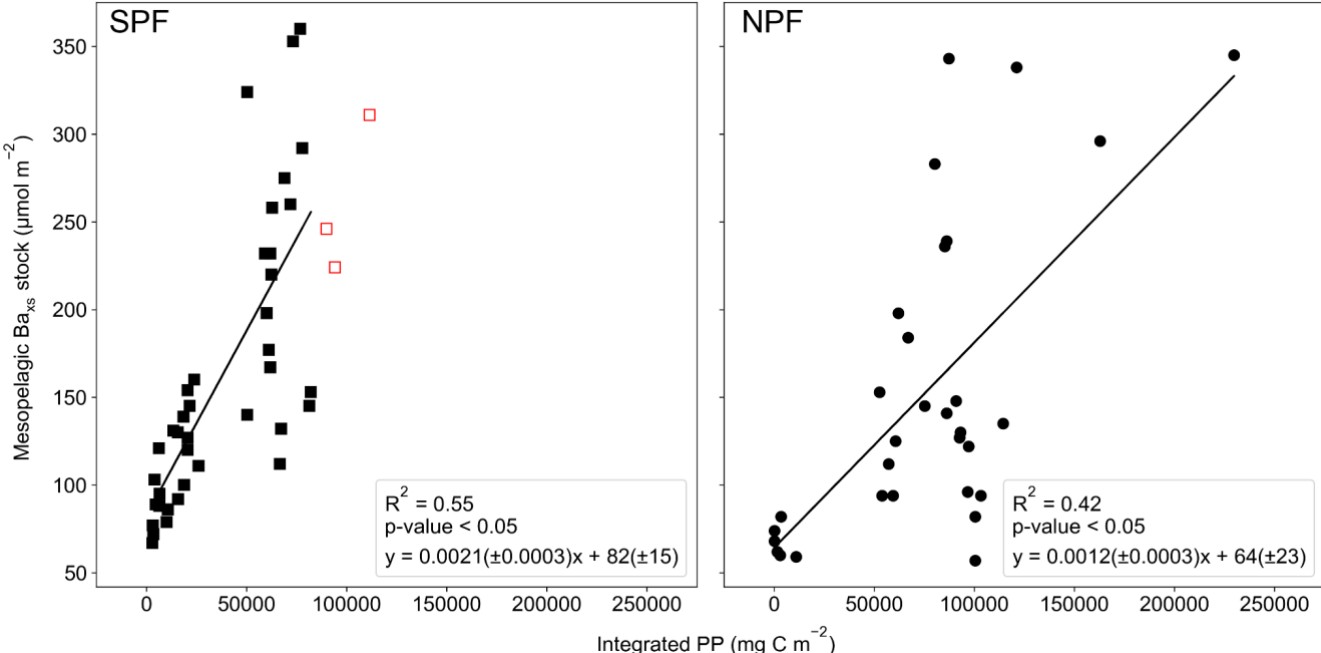

**Figure 5: Integrated mesopelagic $Ba_{xs}$ stock plotted against integrated remotely sensed PP from the preceding September up to one**
**month prior to sampling, all available literature data and winter data from this study, (a) South of the PF (SPF, black squares) and**
**(b) North of the PF (NPF, black circles). Red open squares are data points from our winter dataset where there was not sufficient**
**remote sensing PP data to integrate up to 1 month prior to sampling and available data up to 3 months prior to sampling was plotted**
**but not included in the statistical analysis.**
**4.3    Environmental factors influencing mesopelagic remineralisation and carbon export efficiency**
Estimated POC remineralisation fluxes along the transect (6 - 96 mg C m$^{-2}$ d$^{-1}$) were on the upper end of the range of fluxes
reported in previous studies, with the exception of the STZ station, but within the same order of magnitude for the SO as
estimated from spring to autumn (0.2 - 118 mg C m$^{-2}$ d$^{-1}$; Table S3; Cardinal et al., 2005; Jacquet et al., 2011, 2015; Planchon
et al., 2013). As the bloom season progresses, more efficient remineralisation rates have been reported in multiple studies
(Cardinal et al., 2005; Jacquet et al., 2011; Planchon et al., 2013). However, during late summer as the bloom declines,
observations indicate an inefficient BCP due to enhanced surface nutrient recycling (Dehairs et al., 1992; Jacquet et al., 2011;
Planchon et al., 2013), leading to a decrease in surface POC export (Planchon et al., 2013). Seasonal variation is reported to
be more pronounced northwards within the SO with the least variation observed in the southern Antarctic circumpolar current
(Dehairs et al., 1997; Planchon et al., 2013).

The percentage of mesopelagic POC remineralisation as calculated from estimated POC remineralisation fluxes over integrated remotely sensed PP, for the SO compilation dataset (SPF; $19 \pm 15$ %, n = 39 and NPF; $10 \pm 10$ %, n = 29; mean $\pm$ SD; t-statistic = 2.75; p-value <0.05; Table S3), was ~ 2 fold higher SPF than NPF, revealing the higher surface carbon export efficiency SPF. Our estimates of % POC remineralised fall within the range of reported export efficiencies throughout the SO (2 - 58 %; Jacquet et al., 2011; Morris et al., 2007; Savoye et al., 2008). Our values also support the inverse relationship between export efficiency and productivity, with higher export efficiency in areas of lower production, High Productivity Low E-ratio (HPLE), where E-ratio refers to the ratio between export production and net primary productivity (Fan et al., 2020; Maiti et al., 2013). Estimated mesopelagic POC remineralisation has been reported to account for a significant fraction of exported carbon in the PFZ and southwards, from 31 to 97 %, from spring to summer, whereas it only accounts for ~ 50% in the SAZ and SAF, during summer (Cardinal et al., 2005). A combination of variables can influence surface export efficiency and the magnitude of the subsequent mesopelagic remineralisation, even more so when considering longer timescales. These variables include physical dynamics and interlinked biogeochemical factors, i.e., bacterial activity, phytoplankton community structure, zooplankton grazing and nutrient availability (Bopp et al., 2013; Buesseler and Boyd, 2009; Cardinal et al., 2005; Jacquet et al., 2008b; Pyle et al., 2018). In previous studies, supply and loss via physical transport has been deemed negligible relative to decay and loss via production, due to minimal advection and diffusion gradients observed on the timescale of days to weeks. These processes were therefore assumed to have minimal impact on the mesopelagic signal (Dehairs et al., 1997; Planchon et al., 2013; Rutgers van der Loeff et al., 2011). It has, however, been observed that features such as mesoscale eddies can have an effect on $Ba_{xs}$ distribution by influencing particle patterns on a broad spatial scale, homogenising mesopelagic remineralisation signals by causing relatively flat profiles or shallower remineralisation peaks (Buesseler et al., 2005; Jacquet et al., 2008b). The region of our winter study is known for being a mesoscale eddy hotspot due to the South-West Indian Ridge (Ansorge et al., 2015). In the STZ, extremely dynamic submesoscale activity due to the Agulhas return current may indeed have significantly influenced the mesopelagic signal, and may help explain the absence of correlation with integrated surface PP. On the contrary, south of the STF, the significant correlations seem to indicate that physical transport variability is not the main process affecting the mesopelagic $Ba_{xs}$ signal, and that biogeochemical factors may be dominant.

The Fe-limited SAZ (Ryan-Keogh et al., 2018) and AZ (Viljoen et al., 2018) have generally mixed and seasonally changing assemblages of pico-, nano- and micro-phytoplankton (Eriksen et al., 2018; Gall et al., 2001). Diatoms tend to dominate in the silicate-rich waters south of the PF (Petrou et al., 2016; Rembauville et al., 2017; Wright et al., 2010), whilst seasonally silicate-limited waters north of the PF, favour smaller phytoplankton groups (Freeman et al., 2018; Nissen et al., 2018; Trull et al., 2018). HPLE regimes are indeed characteristic of large areas of the SAZ, mainly due to surface POC accumulation caused by non-sinking particles, tending towards less efficient export of smaller cells (Fan et al., 2020). Even when large particles are abundant in HPLE surface layers, a complex grazing community may prevent the export of large particles (Dehairs et al., 1992; Lam and Bishop, 2007). This can explain the higher surface carbon export efficiency that we estimate SPF compared to NPF. Export efficiency has also been linked to bacterial productivity with efficient surface remineralisation limiting surface POC export, when most of the water column integrated bacterial productivity is restricted to the upper mixed layer (Dehairs et al.,

1992; Jacquet et al., 2011), which can be the case to varying degrees throughout the SO. In the STZ phytoplankton communities
are reported to be dominated by prokaryotic picoplankton including cyanobacteria and prochlorophytes (Mendes et al., 2015).
These groups utilise regenerated nutrients in the surface mixed layer tending towards diminished surface export efficiency
with high concentrations of non-sinking POC (Fan et al., 2020; Planchon et al. 2013). In addition to this, the potential influence
of high submesoscale activity, may explain the low mesopelagic $Ba_{xs}$ measured at the STZ station of this study, despite it being
the station with the highest integrated PP (Figure S1). Linking temporally integrated remotely sensed PP to mesopelagic $Ba_{xs}$
stock, coupled with the added influence of physical dynamics affecting surface export efficiencies, along longer timescales,
could give better estimates of export and remineralisation signals throughout the SO, on an annual and basin scale. Our
estimates of percentage remineralised POC over remotely sensed PP may contribute to the improved modelling of the C cycle
over the SO, on an annual timescale.

## 5   Conclusions

Our unique early winter $Ba_{xs}$ data were similar in magnitude and exhibited the same relationship with $\sigma_\theta$ and dissolved $O_2$
gradients as observed in summer, indicating that processes controlling this signal in summer are still driving the signal in early
winter. The expected decline of the mesopelagic $Ba_{xs}$ signal to background values during winter was not observed in this study,
supporting the hypothesis that this remineralisation proxy likely has a longer timescale than previously reported. The absolute
decline might be delayed due to the cumulative behaviour of mesopelagic $Ba_{xs}$, ongoing remineralisation and barite
precipitation. The "true" SO background value may thus have to be measured at the very end of winter, prior to bloom initiation.
Significant positive correlations north and south of the PF, between mesopelagic $Ba_{xs}$ stock and remotely sensed PP, integrated
from September to 1 month before sampling (Sept - T1), in combination with significant $Ba_{xs}$ accumulation trends obtained
for the SO compilation dataset, suggest an annual timescale. They may also indicate that physical processes do not dominate
the mesopelagic signal on an annual scale, within the SO, and that biogeochemical factors are dominant. There is no significant
difference in mesopelagic $Ba_{xs}$ and POC remineralisation, north and south of the PF, but the significantly higher integrated
remotely sensed PP to the north when compared to the south, indicates a greater export efficiency south of the PF. This is in
accordance with the phenomenon of HPLE regimes which are common throughout the SO, moreso north of the PF than south
of the PF (Fan et al., 2020). The longer timescale of $Ba_{xs}$ and the cumulative behaviour of this proxy in the mesopelagic zone
make it possible to use $Ba_{xs}$ on an annual scale for the estimation of POC remineralisation fluxes throughout the SO and to
better understand how variable environmental factors influence these processes on a basin scale. We believe that the
significance of these relationships will improve as more data become available (e.g., GEOTRACES IDP2021), which will
assist in better understanding and constraining the timescale of remineralisation and carbon export efficiency throughout the
SO.

## 6    Author contribution

This study was conceptualised by N.R.vH, H.P, G.S and E.B. Formal analysis, investigation and validation of data was carried out by N.R.vH, H.P, G.S, E.B and T.J.R-K. N.R.vH and T.J.R-K contributed towards the visualisation of the data. H.P, G.S, T.N.M, A.R, N.L. and E.B contributed towards supervision and resources. Funding was acquired by N.R.vH, T.N.M, A.R and E.B. All authors contributed towards writing, reviewing, and editing of the final manuscript.

## 7    Acknowledgments

This work was supported by the ISblue project, Interdisciplinary graduate school for the blue planet (ANR-17-EURE-0015) and co-funded by a grant from the French government under the program "Investissements d'Avenir". International collaboration was made possible by funding received by the French-South African National Research Foundation (NRF) Collaboration (PROTEA; FSTR180418322331), NRF funding (SNA170518231343 and UID 110715) including funding from South African Department of Science and Innovation, French Ministry of National Education, Higher Education and Research, and the French Ministry of Foreign Affairs and International Development. We would like to thank the captain and crew of the R/V *SA Agulhas II* for their invaluable efforts, as well as all the research participants who assisted our fieldwork. Thanks to Prof. I. Ansorge, Dr M. du Plessis and Dr E. Portela for their assistance with water mass identification, and Dr C. Jeandel for her invaluable expert insight. Finally, we thank the three reviewers, Prof. F. Dehairs, Dr S. Jacquet, and Prof. J. Bishop, for their insightful reviews, thereby improving the quality of our manuscript. A special thank you to Prof. F. Dehairs for sharing the Indigo 3, EPOS 2 and ANTX/6 data.

## 8    Data availability

Data used in this study have been published in the online open-source repository Zenodo and can be accessed at https://doi.org/10.5281/zenodo.6583338 (van Horsten et al., 2022).

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
