# Peer review of "Early winter barium excess in the Southern Indian Ocean as an annual remineralisation proxy (GEOTRACES GIPr07 cruise)"

_Biogeosciences, 2021_

## Referee Comment (RC3)

*Productivity of the Ocean: Present and Past*
eds. W.H. Berger, V.S. Smetacek and G. Wefer, pp. 117–137
John Wiley & Sons Limited
© S. Bernhard, Dahlem Konferenzen, 1989

**Regional Extremes in Particulate Matter Composition and Flux: Effects on the Chemistry of the Ocean Interior**

J.K.B. Bishop

*Lamont-Doherty Geological Observatory*
*Columbia University*
*Palisades, NY 10964, U.S.A.*

*Abstract.* This paper reviews what is known about productivity patterns and vertical flux of organic carbon, opal, and carbonate in the biologically active upper 1000–2000 m of the ocean. Our ability to model particulate carbon flux as a function of depth and primary productivity is much worse than our ability to model particulate carbon flux and regeneration rates as a function of depth once the particulate carbon flux at 100 m is known. A major term missing in equations linking primary production and particle flux appears to be one describing the consumption of particles by macrozooplankton and other large animals.

Simple empirically derived rules based on abundances and chemistry of suspended matter and the mean temperature of the upper 200 m appear sufficient to map the first-order global patterns of production of organic carbon, carbonate, opal, and barium. These may be applicable to both past and present oceanographic conditions.

**INTRODUCTION**

The vertical distribution of carbon species in the ocean is controlled mainly by biologically mediated processes—photosynthesis, feeding, respiration, and decay—which collectively make up the "biological pump" (Fig. 1). The pump intake is located in surface waters where carbon is transformed from dissolved inorganic forms to particulate organic matter by phytoplankton in the presence of light and nutrients. The impeller—or active mechanism in the pump—is the feeding activity of zooplankton and fish which consume the phytoplankton-produced carbon and package a fraction of it in fecal material which sinks into the deep sea at hundreds of meters per day. Daily vertical migrations of zooplankton and nekton may be an additional way for carbon to be transported from the surface to the deep sea (Angel, this

[Figure]

*Fig.* 1—Schematic representation of the biological pump (A) and of physical processes controlling the biological pump (B).

volume). In some oceanic regions, sinking aggregations of phytoplankton may also contribute to the flux. The discharge of the pump occurs deeper where the organic matter in fecal material, animal tissue, or excreted dissolved organic matter (DOM) is oxidized back to dissolved inorganic carbon species (largely through the activities of bacteria and microzooplankton). The pump forms a major pathway for carbon to move from surface waters into the deep sea. The other pathway for carbon to enter the deep ocean is in polar regions where surface waters are subducted to form deep water. Therefore, the operation and efficiency of the "biological pump" may be an important factor in the removal of anthropogenic $CO_2$ from the atmosphere to the ocean. The pump is of importance not only to the global carbon cycle but to the cycles of many other chemical species (Ca, Si, N, P, and trace metals such as Ba, Cd, Ni, Zn, and others).

Operation of the pump depends on two dominant processes: (a) photosynthesis, which depends on light and nutrient availability in the near surface waters, and (b) zooplankton feeding activity. At the base of the euphotic zone, the downward flux (or export flux) of organic matter must, on the whole, be balanced by the upward supply of nutrients (particularly $NO_3^-$). This is the familiar concept of "new production" (Dugdale and Goering 1967; Eppley, this volume). The fraction of photosynthesis supported by $NH_4^+$ and urea is recycled within the euphotic zone. A frequently used assumption is that particle flux through the base of the euphotic zone is the sole balancing term for new production and that this balance is achieved on time scales of days. In some oceanic regions, active transport by migrating animals and deep convective mixing may contribute significantly to the flux of organic matter through the base of the euphotic zone, and alter the time scales over which new production is in balance with export flux.

A schematic view of the biological pump and the physical and biological processes which govern its operation is represented in Fig. 1. The global field of photosynthetically active radiation (PAR, about 50% of incident solar radiation)—the primary driving force for the biological pump—varies with latitude, season, and cloudiness. Nutrient supply to the euphotic zone is governed by turbulent, convective, and advective motions of the water column, which are also predominantly forced from above, and by the nutrient concentrations in subsurface waters. Significant effort has been invested in the development of detailed models incorporating the effects of biological and physical processes on the biological pump (e.g., Kamykowski 1987; Toggweiler, this volume) and, in spite of significant progress, they are still in the developmental stages.

The purpose of this paper is to review some of what is known and not known about the biological pump in the upper 1000–2000 m and to test and suggest some simple relationships which can be used as a basis for modeling the cycles of carbon, Si, alkalinity, and Ba—all of which have well-

**AVERAGE PRIMARY PRODUCTIVITY (LIT.)**

[Figure]

*Fig. 2*—Average carbon primary productivity of the ocean as compiled from the literature (from Berger et al. 1987).

characterized dissolved distributions in the world's ocean. Some derived relationships may appear to be gross extrapolations but they may permit us to understand the first-order patterns of ocean productivity of these elements past and present. We are presently very far from the goal of understanding the effects of sinking and suspended matter on the chemistry of the ocean interior.

**CARBON**

**Global Patterns of Primary Production**

Much of our present understanding of global primary production (the feed for the biological pump) has relied on measurements of primary production measured by the $^{14}C$ incubation technique and on larger data sets describing the nutrient fields in the upper water column (e.g., Koblentz-Mishke et al. 1970). Figure 2 is a map of mean annual primary productivity constructed by Berger et al. (1987). Large gaps in station coverage exist south of 30°S in all oceans and surprisingly few measurements (1-10 per 10 degree square) were available for large regions of the North Atlantic. Nevertheless, Berger and co-workers constructed a set of simple rules relating primary production to available nutrients ($PO_4^{3-}$ at 100 m), light (latitude), and proximity to continental margins which enabled them to fill in the gaps of station coverage. Whether or not their maps are accurate in detail, they are certainly valuable for understanding patterns of carbon fixation in the ocean and can serve as a starting point for estimates of organic matter delivery to the deep sea. Particulate matter residence times based on $^{234}Th/^{238}U$ are highly correlated with values of primary productivity derived largely from old $^{14}C$ data and with particle flux rates estimated using sediment traps (Bruland and Coale 1986). The calculated residence times are not as well correlated with contemporary measurements of primary productivity (Bruland, personal communication). Consequently, the patterns shown in the figure may more closely represent that of export production.

**Parameterizations of Vertical Carbon Flux**

Berger et al. (1987) reviewed sediment trap data published prior to 1986 and various formulations which might be used to model particulate organic carbon flux as a function of depth using maps of primary production. Approximately twice as much data are available now (211 points), mostly for the upper 2000 m from estimates of vertical carbon transport determined by short-term deployments of drifting sediment traps (e.g., Martin et al. 1987) and by large volume *in situ* filtration (LVFS; e.g., Bishop et al. 1987). The new data mostly come from the eastern Pacific but some come from

the N.W. Atlantic, including environments influenced by deep convective mixing, persistent upwelling, and late season destratification. The problems associated with sediment trap and LVFS estimates of vertical flux will not be reviewed here since they are adequately described in the literature. The point of this exercise is to show that empirically derived relationships for vertical carbon flux as a function of primary productivity and depth show errors greatly exceeding the factor of 2–3 uncertainty of the data. Significant improvement may be possible by adding a zooplankton consumer term.

Eight empirical relationships describing particle flux (J: units g C m$^{-2}$ y$^{-1}$) are summarized by Equations 1–8 and are evaluated in Figs. 3a and b.

$$J = PP/(0.024z + 0.21) \quad \text{Suess (1980)} \tag{1}$$

$$J = 0.409PP^{1.41}/z^{0.628} \quad \text{Betzer, Showers et al. (1984)} \tag{2}$$

$$J = 20PP/z \quad \text{Berger et al. (1987), simple fit to data} <1000 \text{ m} \tag{3}$$

$$J = 6.3PP/z^{0.8} \quad \text{Berger et al. (1987), best fit to data} <1000 \text{ m} \tag{4}$$

$$J = 17PP/z + PP/100 \quad \text{Berger et al. (1987), refractory carbon model} \tag{5}$$

$$J = 9PP/z + 0.7PP/z^{0.5} \quad \text{Berger et al. (1987), good fit all data} \tag{6}$$

$$J = 1.286PP/z^{0.734} \quad \text{Pace et al. (1987), vertex: N.E. Pacific} \tag{7}$$

$$J = J_{100}/((z/100)^{0.858}) \quad \text{Martin et al. (1987), vertex: open ocean} \tag{8}$$

The first 7 empirical relationships tested depend only on primary production (PP), and depth (z), and equations 5, 6, and 7 appear least biased and better behaved compared with equations 1–4; however, it is difficult to prefer any one of the three. Martin et al. (1987) showed that if the particle flux at 100 m ($J_{100}$) was known, much better estimates of particle flux to deeper waters could be made. The test of this relationship (Eq. 8) using available data did show significant improvement (Fig. 3b). The major implication is that we must be missing an additional term in the equations which use primary production and depth to predict flux. Simply restated, if we had a global map of particle flux at 100 m, then much better estimates of particle flux and regeneration in deeper waters could be made.

Even using the Martin et al. (1987) relationship, some estimated fluxes fall on the high and low sides of the "measured" flux by as much as an order of magnitude. Martin et al. (1987) showed that the depth exponent used in Eq. 8 was as low as 0.3 for their coastal data and as high as 1.0. If some LVFS data from the Warm Core Rings experiment are included, then the upper limit of the exponent may be as high as 1.5. Is the range of the exponent a consequence of differences between zooplankton feeding demand and particulate matter production in the different environments sampled? The answer is probably yes and evidence to support this is found

[Figure]

*Fig.* 3a—Test of empirical relationships (1–4) for carbon flux as a function of primary production and depth. Small symbols (•) denote carbon flux data from LVFS observations, large symbols (◉) denote carbon flux data from sediment trap observations. The inset is a frequency histogram of the ratio of empirically derived carbon flux to measured carbon flux on a log scale from 0.01 to 100. The y axis of the inset is the number of observations on a scale of 0–30.

in Bishop et al. (1987), who found a consistent relationship between particle flux gradient and zooplankton biomass in the Panama Basin. Unfortunately, few data on distributions of zooplankton biomass have been gathered in conjunction with particle flux studies. Simple rules governing the global distributions of zooplankton (and fish) and their impact on particle flux are needed.

[Figure]

*Fig. 3b*—Test of empirical relationships (5–7) for carbon flux as a function of primary productivity and depth. Also a test of relationship (8) relating particle flux to known particle flux at 100 m depth. It is clear that knowing the particle flux at 100 m allows a much improved estimation of deeper data at the same station than using primary productivity. One would naturally expect this since flux at one depth is used to calculate flux at greater depths. The y axis scales for the inset are 0–30 for all except (8), where the scale is 0–50.

**NITROGEN AND PHOSPHORUS**

No maps of global nitrogen and phosphorus fixation appear to be available. It is reasonable to use typical Redfield ratios for conversion between carbon and nitrogen units. Nitrogen measurements are frequently made along with carbon measurements and no great problems appear to exist. An empirical

relationship similar to Eq. 7 has been derived to predict nitrogen flux as a function of primary production and depth (Pace et al. 1987).

Phosphorus is a problem. Experimental and field evidence shows that phosphorus is preferentially released from particles relative to nitrogen and carbon (e.g., Bishop et al. 1977; Martin et al. 1979). Once below the euphotic zone, the large particles sampled by traps and pumps exhibit C/P ratios greatly exceeding the 106:1 Redfield ratio. Some reported ratios exceeded 1000:1. On the other hand, small particles filtered from seawater at the same depth frequently show ratios much closer to the Redfield ratio (Bishop et al. 1987). Either there is a methodological bias, or another mechanism apart from the vertical transport of phosphorus in large sinking particles gives the ocean its dissolved phosphate distribution.

**SILICA**

**Global Distributions of Silica Productivity**

Although dissolved Si has been extensively measured in the ocean, there have been few attempts to model the global silica cycle. Some still regard Si as having a deep regenerative cycle since profiles of this element frequently show continued increase into the bottom. The work of Berger (1968), Hurd (1972), and Nelson and Goering (1977) provide field and laboratory evidence for shallow silica dissolution. The high deep values for silica in the Pacific may be more related to the fact that the element is dominantly fixed by diatoms living in cold polar and subpolar waters. Such patterns of production are illustrated by Lisitzin (1972; Fig. 4), who used a global primary production map similar to that shown in Fig. 2 and his knowledge of the C/Si ratios of particulate matter filtered continuously from near surface waters during cruises.

**Vertical Fluxes of Silica**

Little or no sediment trap data exists on the Si flux profiles from the euphotic zone to 1000 m. The few sediment trap data published usually include a single sample within this depth interval. LVFS data from the Panama Basin indicate strong Si dissolution in the upper 1000 m.

**CARBONATE**

**Global Estimates of Carbonate Productivity**

According to Lisitzin (1972), the global distributions of coccolith and foraminiferal productivity are maximal between 50°N and 50°S and generally

[Figure]

*Fig.* 4—Annual production of opaline silica in the world ocean (g SiO$_2$ m$^{-2}$y$^{-1}$) (from Lisitzin 1972). Key: 1 = <100; 2 = 100–250; 3 = 250–500; 4 = >500. The cross-hatched area centered at 15°S and 180°W is from a printing error in early primary productivity maps and should be <100 (see Berger et al. 1987, p. 49).

coincide with waters warmer than 10°C. His analysis of suspended carbonate distributions in surface waters showed highest concentrations near the Antarctic convergence in the southern hemisphere from 40–60°S, near the polar front in the northern hemisphere, and along the continental margins. Concentrations in equatorial zones were higher than in the gyres. No maps of global carbonate productivity have been drawn.

**Vertical Fluxes of Carbonate**

There has been no attempt to derive empirical relationships relating vertical fluxes of carbonate to primary productivity. As for silica, few sediment trap data are reported for carbonate flux in shallow waters. The most comprehensive data on fluxes which are published (Betzer, Byrne et al. 1984) appear to be severely biased by entrapment of living pteropods (Harbison and Gilmer 1986). Estimates from LVFS deployments suggest that shallow dissolution of carbonate (especially coccolith carbonate) is a typical occurrence in the eastern equatorial Pacific. This appears to occur in spite of the fact that the water column is supersaturated with respect to calcite from the surface to 1000–2000 m (Bishop et al. 1980, 1987). Evidence for shallow carbonate dissolution in the supersaturated waters of the Atlantic thermocline has been presented by Takahashi et al. (1985). Foraminifera shells have the greatest chance to reach the seafloor unaltered since their residence time in the water column is on the order of days and there is no food value in their empty shells. Foraminifera fluxes may vary by orders of magnitude over time scales of days in productive waters (Bé et al. 1985).

**EMPIRICAL SOURCE FUNCTION MAPS FOR ORGANIC CARBON, OPAL, CARBONATE, AND BARIUM**

The above sections have focused mainly on rate measurements and empirically derived relationships for estimating particulate carbon flux. This section was motivated by the simple rules suggested by Lisitzin (1972, for carbonate and opal) and most recently by Berger et al. (1987, for organic carbon). All suggested that abundances of particulate matter (and hence patterns of production) could be specified by simple transforms between a well measured quantity (ocean temperature, ocean nutrients, light) and the mean abundances of chemical species in suspended particulate matter. This work was also motivated by evidence suggesting that suspended barium concentrations (which frequently showed maxima at several hundred meters) could be used as an index of organic matter regeneration intensity (Bishop 1988). It must be stressed that parts of this section are speculative, but are deliberately so in order to promote discussion.

[Figure]

*Fig.* 5—Average particulate organic carbon, calcium carbonate, opal, and barium in the upper 500 m vs. mean water column temperature at 0–200 m. The (+) symbols denote averages used to derive maps 7–10 from map 6. ◯ = GEOSECS data; ▽ = LVFS data from S.E. Atlantic; △ = LVFS data from N.W. Atlantic.

Data used in this analysis came from three sources: (a) unpublished Atlantic GEOSECS particulate matter data of Spencer and Brewer, (b) LVFS data from the N.W. Atlantic obtained as part of the Warm Core Rings Experiment, and (c) LVFS data from the S.E. Atlantic. Only the 1–53 μm LVFS data are used since the GEOSECS samples (10 liter volumes filtered through 0.4 μm Nucleopore filters) and 1–53 μm LVFS data (up to 25,000 liter volumes of water filtered through 1 μm glass or microquartz fiber filters) have been shown to yield equivalent suspended mass concentrations and the GEOSECS data were biased against >53 μm particles. Without going into detail, each kind of data set was processed to yield an estimate of organic carbon, opal, carbonate, and barium and integrated to 500 m (the 500 m limit was chosen to be deep enough to encompass the subsurface barium maximum) and averages were calculated. The organic carbon and opal calculated from GEOSECS chemical data had to be estimated from iodine measurements and gravimetric differences, respectively.

Temperature is inversely related to nutrient concentration in the upper ocean and therefore can be used as an index of nutrient concentration (Kamykowski and Zentara 1986). Profiles determined at each station were integrated to 200 m and the mean temperature calculated. The rationale for the 200 m limit was to be deep enough to minimize the effects of seasonal warming and cooling and yet to be shallow enough so as to adequately proxy nutrient availability to the euphotic zone. Testing other limits of integration yielded less satisfactory results when compared with the suspended matter data.

Figure 5 shows the results of such a correlation exercise. Barium showed a consistent relationship with temperature both in the GEOSECS data and WCRE data with maximum concentrations at 6°C. Silica increased with decreasing temperature. Superimposed on this increase were high values in some upwelling situations. Calcium carbonate and organic carbon tended to peak at intermediate temperatures. Consistently low concentrations for all elements were found in warmest waters. Means and standard deviations were calculated for the data partitioned in 4 degree temperature intervals.

Annual mean ocean temperature data from Levitus (1982) between 100°W and 20°E were used to calculate 0–200 m average temperatures for the Atlantic Ocean (Fig. 6). The empirical relationship relating the mean concentration and variability (standard deviation) of each element to temperature was used to generate corresponding maps (Figs. 7–10).

One question has been asked about the meaning of such maps constructed from few data points. It was suggested that the high values from upwelling situations might be eliminated since they appear to cause a bias of mapped distributions. The answer is that there is no rationale for eliminating particular points. For example, two data points from the S.E. Atlantic

**ANNUAL MEAN 0-200m TEMPERATURE (°C)**

[Figure]

*Fig.* 6—Annual average 0–200 m ocean temperature (from Levitus et al. 1982).

[Figure]

**O-500m PARTICULATE ORGANIC CARBON**
**(nmol/liter)**

MEAN

VARIABILITY

*Fig.* 7—0–500 m Particulate organic carbon. The left panel displays tne mean concentration derived from ocean temperature (Fig. 6). The right panel is a map of variability (standard deviation) of the data observations about the mean. Lowest concentrations are found in the warmest waters.

**O-500m PARTICULATE BIOGENIC SILICA**
**(nmol/liter)**

MEAN          VARIABILITY

[Figure]

*Fig.* 8—0–500 m Particulate opaline silica. In špite of significant variability, the map of mean silica concentration shows a striking resemblance to the map of silica productivity drawn by Lisitzin (1972). Highest concentrations appear to occur in the subpolar waters south of Iceland and in the Southern Ocean.

[Figure]

**O-500m PARTICULATE CALCIUM CARBONATE**
**(nmol/liter)**

MEAN          VARIABILITY

*Fig.* 9—0–500 m Particulate calcium carbonate. Distributions are highest in waters warmer than 10°C.

**O-500m PARTICULATE BARIUM**
**(pmol/liter)**

MEAN

VARIABILITY

[Figure]

*Fig.* 10—0–500 m Particulate barium. The distributions of this element show the least variability of all variables mapped and Ba appears to be a good indicator of ocean regions with intense organic production.

appear to be anomalous in carbon and calcium but only one is anomalous in silica. One point anomalous in Si was not anomalous in Ba, Ca, or organic carbon. The important point of this exercise is to illustrate that the suspended particulate matter field does correlate with distributions of ocean physical properties and this information can be used to derive first-order patterns for the production function of each element. By considering the variability as well, we can map regions of the ocean which may contain great variability.

The striking feature of these maps is that the features of the carbon, opal, and carbonate fields reflect closely features described in the previous section. Carbon distributions reflect upwelling and ocean frontal regions. For example, productive regions in the eastern equatorial Pacific (Peru upwelling, Costa Rica Dome) are resolved. Carbonate patterns are confined mostly between 50°N and 50°S and are maximal in the subpolar regions consistent with Lisitzin's simple rule. Some variability may occur in polar waters. Opal patterns are strongest in subpolar zones, remain high in polar regions, and correspond closely with Fig. 4. Barium dominates subpolar and upwelling regions, perhaps indicating global patterns of organic matter regeneration intensity. The warm gyres of the North and South Atlantic contain lowest amounts of all quantities.

Maps of variability (with the exception of barium) show maximal values in the subpolar regions and gyre boundaries. These regions may be naturally variable and therefore may be important areas for the transfer of chemically reactive particles into the deep sea. The regions of great variability are regions of the ocean where more work is required to understand the nature of the variability.

The maps illustrate how knowledge of the suspended particle field leads to simple empirical rules for inferring the production patterns of organic matter, carbonate, opal, and barium. Understanding the workings of the biological pump in the subpolar, frontal, and upwelling regions will be needed to fully evaluate patterns of ocean productivity past and present.

**CONCLUSION**

Simple rules may exist allowing us to specify the operation of the biological pump on a global basis. Clearly, there is a need to define more of the rules of operation of the pump through cross-disciplinary analysis of existing data and through new field work.

*Acknowledgements.* The author wishes to thank Drs Derek Spencer and Peter Brewer for providing access to their unpublished GEOSECS Atlantic particulate matter data. Drs W. Berger, P.E. Biscaye, S. Emerson, J. Marra, and V. Ittekkot are thanked for their reviews of the text. Sample collection

and analysis were supported by NSF grants OCE88-02773 and OCE85-13420 and ONR contract N00014-80-C-0098. This effort was supported by the NASA cooperative agreement NCC5-29A to Columbia University. L-DGO contribution number 4385.

**REFERENCES**

Bé, A.W.H.; Bishop, J.K.B.; Sverdlove, M.S.; and Gardner, W.D. 1985. Standing stock, vertical distribution and flux of planktonic foraminifera in the Panama Basin. *Mar. Micropal.* **9**: 307–333.

Berger, W.H. 1968. Radiolarian skeletons: solution at depths. *Science* **159**: 1237–1239.

Berger, W.H.; Fisher, K.; Lai, C.; and Wu. G. 1987. Ocean carbon flux: global maps of primary production and export production. In: Biogeochemical Cycling and Fluxes between the Deep Euphotic Zone and Other Oceanic Realms, ed. C. Agegian. NOAA Symp. Ser. for Undersea Research, NOAA Undersea Research Program, vol. 3(2). Preprint in SIO ref. 87–30.

Betzer, P.R.; Byrne, R.H.; Acker, J.G.; Lewis, C.S.; Jolley, R.R.; and Feely, R.A. 1984. The oceanic carbonate system: a reassessment of biogenic controls. *Science* **226**: 1074–1076.

Betzer, P.R.; Showers, W.J.; Laws, E.A.; Winn, C.D.; DiTullio, G.R.; and Kroopnick, P.M. 1984. Primary productivity and particle fluxes on a transect of the equator at 153 W in the Pacific Ocean. *Deep-Sea Res.* **31**: 1–12.

Bishop, J.K.B. 1988. The barite-opal-organic carbon association in oceanic particulate matter. *Nature* **233**: 241–243.

Bishop, J.K.B.; Collier, R.W.; Ketten, D.R.; and Edmond, J.M. 1980. The chemistry, biology and vertical flux of particulate matter from the upper 1500 m of the Panama Basin. *Deep-Sea Res.* **27**: 615–640.

Bishop, J.K.B.; Edmond, J.M.; Ketten, D.R.; Bacon, M.P.; and Silker, W.B. 1977. The chemistry, biology and vertical flux of particulate matter from the upper 400 m of the equatorial Atlantic ocean. *Deep-Sea Res.* **24**: 511–548.

Bishop, J.K.B.; Stepien, J.C.; and Wiebe, P.H. 1987. Particulate matter distributions, chemistry and flux in the Panama Basin: response to environmental forcing. *Prog. Ocean.* **17**: 1–59.

Bruland, K., and Coale, K.H. 1986. Surface water $^{234}Th/^{238}U$ disequilibria: spatial and temporal variations of scavenging rates within the Pacific Ocean. In: Dynamic Processes in the Chemistry of the Upper Ocean, eds. J.D. Burton, P.G. Brewer, and R. Chesselet, pp. 159–172. New York: Plenum.

Dugdale, R.C., and Goering, J.J. 1967. Uptake of new and regenerated forms of nitrogen in primary productivity. *Limnol. Ocean.* **12**: 196–206.

Harbison, G.R., and Gilmer, R.W. 1986. Effects of animal behavior on sediment trap collections: implications for the calculation of aragonite fluxes. *Deep-Sea Res.* **33**: 1017–1024.

Hurd, D.C. 1972. Factors affecting solution rate of biogenic opal in sea water. *Earth Planet. Sci. Lett.* **15**: 411–417.

Kamykowski, D. 1987. A preliminary biophysical model of the relationship between temperature and plant nutrients in the upper ocean. *Deep-Sea Res.* **34**: 1067–1079.

Kamykowski, D., and Zentara, S.-J. 1986. Predicting plant nutrient concentration from temperature and sigma-t in the world ocean. *Deep-Sea Res.* **33**: 89–105.

Koblentz-Mishke, O.J.; Volkovinsky, V.V.; and Kabanova, J.G. 1970. Plankton primary production of the world ocean. In: Scientific Exploration of the South

Pacific, ed. W.S. Wooster, pp. 183–193. Washington, D.C.: National Academy of Sciences.

Levitus, S. 1982. Climatological Atlas of the World Ocean. NOAA Prof. Paper 13, US Dept of Commerce.

Lisitzin, A.P. 1972. Sedimentation in the World Ocean. Spec. Publ. 17. Tulsa, OK: Soc. Econ. Paleon. Miner.

Martin, J.H.; Knauer, G.A.; and Bruland, K. 1979. Fluxes of particulate carbon, nitrogen, and phosphorus in the upper water column of the northeast Pacific. *Deep-Sea Res.* **26**: 97–108.

Martin, J.H.; Knauer, G.A.; Karl, D.M.; and Broenkow, W.W. 1987. VERTEX: carbon cycling in the northeast Pacific. *Deep-Sea Res.* **34**: 267–286.

Nelson, D.M., and Goering, J.J. 1977. Near-surface silica dissolution in the upwelling region off northwest Africa. *Deep-Sea Res.* **24**: 31–36.

Pace, M.L.; Knauer, G.A.; Karl, D.M.; and Martin, J.M. 1987. Primary production, new production and vertical flux in the eastern Pacific Ocean. *Nature* **325**: 803–804.

Suess, E. 1980. Particulate organic carbon flux in the ocean—surface productivity and oxygen utilization. *Nature* **288**: 260–263.

Takahashi, T.; Broecker, W.S.; and Langer, S. 1985. Redfield ratio based on chemical data from isopycnal surfaces. *J. Geophys. Res.* **90**: 6907–6924.

---

## Author Comment (AC1)

**Comment on bg-2021-42**

Frank Dehairs (Referee)

Referee comment on "Early winter barium excess in the Southern Indian Ocean as an annual remineralisation proxy (GEOTRACES GIPr07 cruise)" by Natasha René van Horsten et al., Biogeosciences Discuss., https://doi.org/10.5194/bg-2021-42-RC1, 2021

This manuscript brings much wanted information about the Southern Ocean particulate biogenic Ba (Baxs) distribution during winter conditions. Previous studies in the S.O. were all conducted during spring to autumn conditions, showing a seasonal progress of the Baxs signal but lacking information on winter conditions when plankton activity is at a minimum. The mesopelagic Baxs inventory is gauged against integrated PP covering the growth period preceding the sampling, and these data are combined with literature data revealing an interesting correlation.

Response: We thank Pr. Frank Dehairs for this very positive comment.

Please see below our responses to the revisions and comments (in blue) and the excerpts from the revised manuscript (in red).

I wonder why authors, when comparing their data with literature, have considered data from specific expeditions and not from all available data for the S.O. In particular the Baxs data presented in Dehairs et al. (GBC 1990; INDIGO 3 expedition, 1987) for the same general area as studied by the present authors were not considered. During the INDIGO 3 several stations were occupied along approx. 30°E between 65°S and 57°S. S to N Baxs inventories are similar to values reported in the present ms., confirming indeed that microbial activity in the mesopelagic area is still ongoing during winter period. Further data that have not been included in the Baxs inventory – PP comparison are those from Dehairs et al. (1997) obtained during Polarstern ANT X/6 expedition along 6°W in early season. If possible, these two data sets should be included in the compilation.

Response: In the initial manuscript, the $Ba_{xs}$ data were plotted against satellite-derived PP, integrated over months prior to sampling. At the time of the INDIGO 3 and ANT X/6 cruises, there was no remotely sensed PP data available and the PP data from Dehairs et al. (1997) were measured during the cruise and are not representative of integrated PP over months preceding the study. Therefore, these data were not initially included in our compilation.

Nevertheless, the mesopelagic $Ba_{xs}$ stock ($\mu mol\ m^{-2}$) is now plotted against day of year sampled (Figure 4b and c, see below) in the revised manuscript, including the INDIGO 3, ANT X/6 and EPOS 2 data (see comment by Stéphanie Jacquet). However, as stated in the Figure caption, these data must be considered with caution because these samples were not digested using HF. This can lead to an underestimation of aluminium concentrations and an overestimation of $Ba_{xs}$, where there are possible significant lithogenic inputs (e.g., close to Antarctica and downstream of the Drake Passage).

In order to investigate this hypothesis, for the first time, we compiled a SO mesopelagic $Ba_{xs}$ stock dataset with all available literature data including data from this study (Figure 4a, Table S3). The mesopelagic $Ba_{xs}$ stock was integrated over the $Ba_{xs}$ peak depth range (as identified in each study). As can be seen on the map of the compilation dataset (Figure 4a), these data points were collected in the three basins of the SO, over 20 years. Despite this diversity, a statistically significant accumulation of mesopelagic $Ba_{xs}$ with time, SPF and NPF (Figure 4b and c) is still observed. Mesopelagic $Ba_{xs}$ accumulates at a rate of 0.86 (±0.15) $\mu mol\ m^{-2}\ d^{-1}$, SPF ($R^2$ = 0.43, p-value < 0.05, n =

43; Figure 4b), and at 0.88 (±0.20) µmol m$^{-2}$ d$^{-1}$, NPF ($R^2$ = 0.41, p-value < 0.05, n = 31; Figure 4c), with no statistically significant difference between the two zones (Welch's t-test = 0.24; p-value = 0.80).

The seasonal signal for PP over the growing season (Figure 4d and e) clearly shows that the highest values occur between January and February (day 125 to 175 of the year), thereafter, steadily decreasing to minimal values in July (~ day 310 of the year, i.e., during our study). The mesopelagic Ba$_{xs}$ accumulation over time can, therefore, not be matched with the remotely sensed PP measured during the month of sampling

[Figure]

**Figure 4: (a) Positions of Ba$_{xs}$ observations compiled from all known SO studies, on a cylindrical equal-area projection of the SO, the three SO basin cut offs are indicated by the dashed black lines, from left to right, Pacific, Atlantic and Indian. Integrated mesopelagic Ba$_{xs}$ stock plotted against day of year sampled, with the 1st of September set as day 1, for all available literature data and winter data from this study. Data was split into two zones using the Polar Front (PF) to divide the SO; (b) South of the PF (SPF) and (c) North of the PF (NPF). Monthly averaged remotely sensed PP plotted against day of year, for locations and dates of the SO compilation dataset and winter data from this study; (d) SPF and (e) NPF. Open circles are data points from studies which did not use HF in the particulate sample digestion procedure, regressions did not include these data, there was, however, no significant difference when including these data points.**

Authors do not provide information on how integrated PP was obtained for the compilation of literature data. Figure 1 should differentiate the different Baxs data sets.

Response: All PP data in the manuscript is satellite derived integrated PP using the CbPM PP algorithm, no in situ PP measurements were taken into consideration as we considered PP prior to the sampling of mesopelagic Ba$_{xs}$.

The methodology of integrated remotely sensed PP is included under section 2.5 of the original manuscript, with further details on subsampling in the discussion, section 4.2. We have expanded on this in the discussion (section 4.2) to state that this methodology was used for all considered studies in the dataset.

A possible link between the integrated mesopelagic Ba$_{xs}$ stock and the corresponding integrated remotely sensed PP was assessed for all studies conducted after September 1997, when remotely sensed PP data became available. To do so, we first estimated that sub millimetre sized aggregates, in which barite crystals produced, would take ~ 20 days to sink down to 1000 m (considered as the bottom of the mesopelagic zone, in this study), using a sinking speed of 50 m d$^{-1}$ that corresponds to an average literature value (50 - 100 m d$^{-1}$: Riebesell et al., 1991 ; 50 - 430 m d$^{-1}$ around South Georgia: Cavan et al. 2015; ~ 100 m d$^{-1}$ in the Southern Ocean as reviewed in Laurenceau-Cornec et al., 2015; 10 - 150 m d$^{-1}$: McDonnell and Buesseler, 2010). Assuming a maximum surface current speed of 0.2 m s$^{-1}$ (Ferrari and Nikurashin, 2010), it was estimated that these aggregates would have originated, 346 km west from the station that was sampled for mesopelagic Ba$_{xs}$, ~ 20 days prior. Using this distance, the dimensions of the sample area were set with the southernmost station (TM1) of this study, where degrees of longitude cover the smallest area. For the sake of consistency this sampling area was applied to all sampling locations of the considered dataset. The integrated remotely sensed PP (see section 2.5) was then averaged spatially, positioned 6° upstream longitudinally, and 1° latitudinally centred around each sampled station, in order to capture the surface PP that is assumed to translate to the mesopelagic remineralisation and Ba$_{xs}$ stock.

An additional map has been included to identify the location of studies in the compilation dataset (Figure 4a).

Authors do not provide any information on sea ice extent relative to position of the southernmost station.

Response: This information is now included in the revised manuscript.

The marginal ice zone, identified as the position of 30% ice cover, was positioned at 61.7°S, approximately 3° (356km) south of the southernmost station (de Jong et al., 2018). Therefore, a potential sea ice influence on our study area can be disregarded.

While it makes sense to compare Baxs inventories with PP intensity in the months preceding the sampling, the coinciding Chlorophyll data shown in Fig. S1 still are relatively elevated reaching about 0.5 µg/L at the PF and in the SAZ, this taking into account that S.O. Chl values of 1 µg/L can be considered bloom values. Please comment.

Response: Whilst we agree with the reviewer that these Chl-α values are high, these measurements were performed by fluorometry. Fluorometric methods of measuring Chl-α are prone to errors, particularly in areas where there are high concentrations of diatoms. This is due to the presence of high Chl-c pigment concentrations in diatoms, and the spectral overlap between Chl-α and Chl-c falsely inflating Chl-α concentrations. This is best demonstrated in the study by Moutier et al. (2018, doi: 10.3390/rs11151793, Table 5) where they show that Chl-α determined by fluorometry can be almost double that as measured by HPLC. Unfortunately, no HPLC samples were collected during this study, so we do not have access to Chl-α concentrations by this method.

The issues with the use of fluorometric Chl-α have been reported by many studies (Pereira et al., 2018; Marrari et al., 2006; Gibbs, 1979; Welschmeyer, 1994; Roesler et al., 2017; Kumari, 2005; Lorenzen, 1981; Trees et al., 1985; Bianchi et al., 1995; Dos Santos et al., 2003). It is for this very reason that NASA only uses HPLC derived Chl-α in the cross-check calibration of remote sensing derived values.

Due to this issue with the Chl-α measurements, we have removed them from the revised manuscript, and we have instead added time series, area-averaged remotely sensed CbPM PP plots for each station sampled, indicating that PP was at a minimum ~ 2 months prior to the time of sampling.

[Figure]

**Figure 3: Time series, area-averaged remotely sensed CbPM-PP (mg C m$^{-2}$ day$^{-1}$), monthly average from 08/2016 to 08/2017, dashed vertical lines indicate sampling date.**

Not sure Figure 4 adds to the understanding. This figure could be omitted.

Response: As the three reviewers agree on this point, we have decided to remove this figure from the revised manuscript.

Specific comments

Line 67, page 3: Dehairs et al. 1980 more appropriate as ref. here than dehairs et al. 1997.

Response: This reference has been modified.

It is defined as the "biogenic" portion of particulate Barium (pBa) as barite crystals, formed by the decay of bio-aggregates below the surface mixed layer (Bishop, 1988; Dehairs et al., 1980; Lam and Bishop, 2007; Legeleux and Reyss, 1996; van Beek et al., 2007).

Lines 58-59: Surface export is set by the deficit (not excess) of 234Th activity vs. 238U activity. Specify that 234/238 ratios >1 can occur below the upper 100m, or so, reflecting remineralisation.

Response: This has been corrected.

Surface export is set by the deficit of $^{234}$Th activities over $^{238}$U activities. When $^{234}$Th/$^{238}$U ratios are larger than 1, below the surface mixed layer, this can reflect remineralisation processes, integrating processes over a 2 to 3 week period (Buesseler et al., 2005; Planchon et al., 2013).

Lines 26-28 page 10 and lines 75-76 page 16: Sample numbers n=39 (SPF) and NPF (n=31) pertain to what? Table S1 shows only data from the present study and not the compilation data set.

Response: The n values were referring to the number of observations in the compilation dataset for which there is remotely sensed PP available, i.e., studies after September 1997, split between two zones, SPF and NPF respectively.

Section 3.4 (lines 26 - 28, page 10) has been removed and this information has been added to the discussion under section 4.3 in the revised manuscript, after the compilation dataset has been properly introduced.

The compilation dataset has also now been included as supplementary Table S3 and in-text references to the supplementary table have been corrected.

Lines 46-47 page 11: Opposite gradients of Baxs and diss. O2. Please provide more detail.

Response: We have updated sentences to provide clarity on the gradients observed on the $Ba_{xs}$ and $O_2$ profiles.

Additionally, decreases observed in dissolved $O_2$ profiles along the transect were also accompanied by coinciding, increases in $Ba_{xs}$, in line with $O_2$ consumption due to remineralisation within the mesopelagic zone (Figure 2) (Cardinal et al., 2005; Jacquet et al., 2005, 2011).

Lines 65-79 page 16: These sentences are confusing. If there is no significant difference in relative amount of POC remineralized relative to PP (all stations except STZ), then there is no difference in response of Ba relative to PP at the different stations.? Only the STZ site behaves differently.

Response: We agree that this section of the discussion was indeed confusing, and therefore we took a closer look at our data and became aware of two outliers, both being much higher than 100%, which completely altered the mean and variance of the % POC remineralised of the NPF data. We have excluded these two outliers from the % POC remineralised data, as they both fell outside the limit of acceptance of three times the standard deviation of the dataset. This results in a significant difference in the % POC remineralised, between the two zones (NPF and SPF). Making a lot more sense, as this highlights the difference in surface export efficiency between the two regimes.

The physics at the time of sampling for the two outliers (Jacquet et al., 2004), reveals that these locations were more affected by physics than was the case for the rest of our dataset. Specifically, the southern edge of a subtropical Tasman Sea eddy, coinciding with the STF, to the north, and a cold core eddy from the SAF to the south, creating a highly dynamic region between the STF and SAF (Jacquet et al., 2004). These physical factors would affect the mesopelagic signal, thereby masking the surface to mesopelagic relationship usually seen when physics is not the dominating process, as was the case for the rest of our dataset.

We thank Pr. Frank Dehairs for bringing this error to our attention. We have rewritten the discussion to rectify this.

The percentage of mesopelagic POC remineralisation as calculated from estimated POC remineralisation fluxes over integrated remote sensing PP and determined for the SO compilation dataset (SPF; 19 ± 15 %, n = 39 and NPF; 10 ± 10 %, n = 29; mean ± SD; t-statistic = 2.75; p-value <0.05; Table S3), was ~ 2 fold higher SPF than NPF, revealing the higher surface carbon export efficiency SPF.

Also, high productivity, low export can be associated with large particles in the surface layer (see Lam & Bishop, 2007). High surface water productivity associated with low export has also been described in Jacquet, Lam, Trull, Dehairs, DSR II, 2011. The possibility that high phyto biomass attracts more grazing and is more depending on recycled production (NH4 based) and thus results in smaller export (more surface water recycling) and possibly lower mesopelagic Ba, is reported also in Dehairs et al., 1992.

Response: The discussion on HPLE regions has been expanded to include this information.

HPLE regimes (High Productivity Low E-ratio, e-ratio referring to the ratio between export production and net primary productivity, Fan et al., 2020) are indeed characteristic of large areas of the SAZ. They are mainly due to surface POC accumulation caused by non-sinking particles, tending towards less efficient export of smaller cells (Fan et al., 2020). Even when large particles are abundant in HPLE surface layers, a complex grazing community may prevent the export of large particles (Dehairs et al., 1992; Lam and Bishop, 2007). This can explain the higher surface carbon export efficiency that we estimate in the AZ compared to the SAZ. Export efficiency has also been linked to bacterial productivity, when most of the water column integrated bacterial productivity is restricted to the upper mixed layer, efficient surface remineralisation limits surface POC export (Dehairs et al., 1992; Jacquet et al., 2011).

Line 76 page 16: ".. are comparable to surface export efficiency obs. In this region ..." which region?

Response: We were referring to the SO. The sentence has been amended to clearly state that.

Line 88 page 17: this sentence is unclear. Similar latitudinal trend (of what?); higher values NPF (values of what?)

Response: We were referring to the latitudinal trend of mesopelagic $Ba_{xs}$ concentrations. The sentences have, however, been removed from the revised manuscript.

Line 90 page 17: saturated vs undersaturated: specify saturation for dissolved Ba.

Response: Unfortunately, there was no dBa measured during our study. Any mention of Ba saturation has been removed from the conclusion as this was not specifically discussed in our manuscript.

Table S2: Add lat. position for each site

Response: This has been added to Table S2.

Figure S1: top panel indicate AZ, PFZ, SAZ. Why is the STZ station not reproduced here?

Response: We have overlaid the zones on the figure. The STZ station was included at 41°S.

[Figure]

**Figure S1: Top panel is the latitudinal trend, south to north, left to right, of winter integrated mesopelagic Ba$_{xs}$ stock concentrations (black triangles). The bottom panel is the latitudinal trend of the corresponding integrated annual remote sensing PP (black circles). Sampling zones are overlaid in grey, namely SPF, NPF and STZ.**

Table S3: Add the Lat-Long range for the basin regions

Response: This has been added to Table S3, which has also been updated to include all data pertaining to the compilation dataset.

---

## Author Comment (AC2)

**Comment on bg-2021-42**

Stéphanie Jacquet (Referee)

Referee comment on "Early winter barium excess in the Southern Indian Ocean as an annual remineralisation proxy (GEOTRACES GIPr07 cruise)" by Natasha René van Horsten et al., Biogeosciences Discuss., https://doi.org/10.5194/bg-2021-42-RC2, 2021

This manuscript presents a new data set of excess particulate barium (Baxs) concentrations in the Southern Ocean during winter conditions. Correlation with integrated PP and data from literature is interesting. My major comment concerns the conclusion that the Ba proxy would have a longer timescale than previously thought. I don't think that there is a cumulative effect on the Baxs signal for the reasons explained below. I suggest that authors revise their discussion (and reformulate abstract & conclusion).

Response: we thank Dr Stéphanie Jacquet for the review of our manuscript.

Please see below our responses to the revisions and comments (in blue) and the excerpts from the revised manuscript (in red).

The hypothesis of this study, that the timescale of $Ba_{xs}$ may be longer than a few days to weeks, has already been suggested by earlier studies, e.g.:

"Significant carry over of Bap between successive plankton growth seasons might occur" (Dehairs et al., 1997)

"The time-delay required to build-up the meso-$Ba_{xs}$ (few weeks) supports the fact that this proxy is not a snapshot (on the contrary of N-uptake for instance) but rather a monthly average of remineralisation." (Cardinal et al., 2005)

We hope that our revised manuscript will be more convincing in this regard. Figure 3 (see below) will be added to the revised manuscript, showing that PP was at a minimum ~ 2 months prior to the time of our sampling. We have also added Figure 4 (see below), showing significant increases of mesopelagic $Ba_{xs}$ from September to July, for all data compiled from the different basins of the Southern Ocean.

Chl a data reported in Figure S1 indicate that stations (northern 50°S) experienced production (even of low intensity). This should be compared with Chl a and Baxs data from other campaigns (e.g., Cardinal et al., 2005; Blain et al., 2007). The winter period appears to be productive in this sector. This would explain why Baxs present similar contents as reported during other seasons.

Response: Whilst we agree with the reviewer that these Chl-α values are high, these measurements were performed by fluorometry. Fluorometric methods of measuring Chl-α are prone to errors, particularly in areas where there are high concentrations of diatoms. This is due to the presence of high Chl-c pigment concentrations in diatoms, and the spectral overlap between Chl-α and Chl-c falsely inflating Chl-α concentrations. This is best demonstrated in the study by Moutier et al. (2018, doi: 10.3390/rs11151793, Table 5) where they show that Chl-α determined by fluorometry can be almost double that as measured by HPLC. Unfortunately, no HPLC samples were collected during this study, so we do not have access to Chl-α concentrations by this method.

The issues with the use of fluorometric Chl-α have been reported by many studies (Pereira et al., 2018; Marrari et al., 2006; Gibbs, 1979; Welschmeyer, 1994; Roesler et al., 2017; Kumari, 2005; Lorenzen, 1981; Trees et al., 1985; Bianchi et al., 1995; Dos Santos et al., 2003). It is for this very reason that NASA only uses HPLC derived Chl-α in the cross-check calibration of remote sensing derived values.

Due to this issue with the Chl-α measurements, we have removed them from the revised manuscript, and we have instead added time series, area-averaged remotely sensed CbPM PP plots for each station sampled, indicating that PP was at a minimum ~ 2 months prior to the time of sampling.

[Figure]

**Figure 3: Time series, area-averaged remotely sensed CbPM-PP (mg C m$^{-2}$ day$^{-1}$), monthly average from 08/2016 to 08/2017, dashed vertical lines indicate sampling date.**

Data should be compared to results from the Indigo3, EPO2 and ANTX/6 cruises.

Response: In the initial manuscript, the Ba$_{xs}$ data were plotted against satellite-derived PP, integrated over months prior to sampling. At the time of the INDIGO 3 and ANT X/6 cruises, there was no remotely sensed PP data available and the PP data from Dehairs et al. (1997) were measured during the cruise and are not representative of integrated PP over months preceding the study. Therefore, these data were not initially included in our compilation.

Nevertheless, the mesopelagic Ba$_{xs}$ stock (µmol m$^{-2}$) is now plotted against day of year sampled (Figure 4b and c, see below) in the revised manuscript, including the INDIGO 3, ANT X/6 and EPOS 2 data. However, as stated in the Figure caption, these data must be considered with caution because these samples were not digested using HF. This can lead to an underestimation of aluminium concentrations and an overestimation of Ba$_{xs}$, where there are possible significant lithogenic inputs (e.g., close to Antarctica and downstream of the Drake Passage).

In order to investigate this hypothesis, for the first time, we compiled a SO mesopelagic Ba$_{xs}$ stock dataset with all available literature data including data from this study (Figure 4a, Table S3). The mesopelagic Ba$_{xs}$ stock was integrated over the Ba$_{xs}$ peak depth range (as identified in each study). As can be seen on the map of the compilation dataset (Figure 4a), these data points were collected in the three basins of the SO, over 20 years. Despite this diversity, a statistically significant accumulation of mesopelagic Ba$_{xs}$ with time, SPF and NPF (Figure 4b and c) is still observed. Mesopelagic Ba$_{xs}$ accumulates at a rate of 0.86 (±0.15) µmol m$^{-2}$ d$^{-1}$, SPF ($R^2$ = 0.43, p-value < 0.05, n = 43; Figure 4b), and at 0.88 (±0.20) µmol m$^{-2}$ d$^{-1}$, NPF ($R^2$ = 0.41, p-value < 0.05, n = 31; Figure 4c), with no

statistically significant difference between the two zones (Welch's t-test = 0.24; p-value = 0.80).

The seasonal signal for PP over the growing season (Figure 4d and e) clearly shows that the highest values occur between January and February (day 125 to 175 of the year), thereafter, steadily decreasing to minimal values in July (~ day 310 of the year, i.e., during our study). The mesopelagic $Ba_{xs}$ accumulation over time can, therefore, not be matched with the remotely sensed PP measured during the month of sampling.

[Figure]

**Figure 4: (a) Positions of $Ba_{xs}$ observations compiled from all known SO studies, on a cylindrical equal-area projection of the SO, the three SO basin cut offs are indicated by the dashed black lines, from left to right, Pacific, Atlantic and Indian. Integrated mesopelagic $Ba_{xs}$ stock plotted against day of year sampled, with the 1st of September set as day 1, for all available literature data and winter data from this study. Data was split into two zones using the Polar Front (PF) to divide the SO; (b) South of the PF (SPF) and (c) North of the PF (NPF). Monthly averaged remotely sensed PP plotted against day of year, for locations and dates of the SO compilation dataset and winter data from this study; (d) SPF and (e) NPF. Open circles are data points from studies which did not use HF in the particulate sample digestion procedure, regressions did not include these data, there was, however, no significant difference when including these data points.**

Line 15-19: Please revise the abstract (and part of the conclusion). I don't think it's a question of timescale and cumulative effect. If POC is produced in surface and that remineralization is sustained at mesopelagic depths, Baxs will be produced, independently from the season. There is no clues that POC material could accumulate at mesopelagic depths and conducts to latter (weeks to months after the growth season) remineralization and Baxs

Response: Indeed, we agree that POC may not accumulate in the mesopelagic zone. We are referring to the accumulation of $Ba_{xs}$ in the mesopelagic zone. The barite crystals released in the mesopelagic zone due to remineralisation of POC can become suspended due to its low solubility and slow sinking speeds of ~0.3 m d$^{-1}$ (Sternberg et al., 2008; review comment by Prof. J. Bishop). It has been found that only up to 30% of marine barite could reach the ocean floor due to its low solubility and packaging in fast settling fecal pellets (Paytan and Kastner, 1996). This leaves a minimum of 70% which does not

settle out and will become suspended in the mesopelagic zone, where dissolution and reaggregation would be the main processes controlling the concentration of $Ba_{xs}$, thereby suggesting accumulation in the ocean interior as more $Ba_{xs}$ is exported from the surface ocean. We have clarified this in the revised manuscript.

A compilation of available SO mesopelagic $Ba_{xs}$ data, including ours, shows a mesopelagic $Ba_{xs}$ accumulation from September to July that correlates with integrated remotely sensed primary productivity (PP), suggesting a possible annual timescale for this proxy.

The integrated mesopelagic $Ba_{xs}$ stock ($\mu mol\ m^{-2}$) over the mesopelagic layer (100 - 1000 m) was calculated from the DWA $Ba_{xs}$ in order to investigate the link between the accumulated $Ba_{xs}$ mesopelagic signal and the corresponding integrated remote sensing primary productivity (PP).

As can be seen on the map of the compilation dataset (Figure 4a), these data points were collected in the three basins of the SO, over 20 years, and a statistically significant accumulation of mesopelagic $Ba_{xs}$ with time, SPF and NPF (Figure 4b and c) is still observed.

Figure 4: not necessary -it does not add to the understanding. It should be (in-depth) compared to contrasts reported in Jacquet et al. (2011; SAZ-SENSE cruise): diatoms vs. flagellate, PP, EP, Fe depletion or enrichment, type of aggregates. The effect of the contrasts on Baxs and remineralization during SAZ-SENSE was opposite to these reported during KEOPS (Jacquet et al., 2008) and EIFEX (Jacquet et al., 2008) cruises. This should be compared to the present data set.

Response: As the three reviewers agree on this point, we have decided to remove this figure from the revised manuscript.

Line 71-82 p11-12 (and Line 90 p17: not clear, please reformulate). Are dissolved Ba and SI available? The SO is globally undersaturated (SI<0.9) or at the equilibrium (0.9<SI<1.1) with respect to barite. Saturation is unusual. Please correct it line 73. Also, in productive situations (and deep POC transfer), it is common that Baxs at 1000 m depths remains larger than the "180 pM" SO reference.

Response: Unfortunately, dissolved Ba was not measured during our study, therefore the SI could not be calculated. We have removed any mention of barite saturation in the revised manuscript as we do not specifically address this topic.

We agree that $Ba_{xs}$ at 1000m can be higher than the SO $Ba_{residual}$ concentration of 180pM due to deep POC transfer, this is the reason for us only using concentrations from depths below 2000m, for the consideration of $Ba_{residual}$ concentrations (line 52, page 6).

Finally, as recently reported in Jacquet et al. (https://doi.org/10.5194/bg-2020-271; Peacetime cruise) remineralization at mesopelagic depths could be restricted to the upper mesopelagic layer or extend up to 1000 m depending the system functioning during a same season. This leads to major differences in the Baxs background reached at 1000 m depths.

Response: We agree fully with this comment and in the paragraph starting on line 51 page 11, we refer to remineralisation and deeper $Ba_{xs}$ peaks NPF due to deep POC export.

Ba$_{xs}$ profiles exhibited similar distributions to those reported throughout bloom seasons in the SO, with distinct peaks observed within the mesopelagic zone across all stations. Earlier in the bloom season peaks mostly occur within the upper half of the mesopelagic zone (100 - 500 m) (Cardinal et al., 2001, 2005; Jacquet et al., 2005, 2008a, 2011, 2015), but as the season progresses, they deepen down towards the bottom half of the mesopelagic zone (500 - >1000 m) (Jacquet et al., 2008b, Planchon et al., 2013). Deepening and widening of the remineralisation depth range can be expected as the season progresses, due to continued remineralisation taking place as particles sink to the bottom of the mesopelagic zone (Lemaitre et al., 2018; Planchon et al., 2013). This is also what we observed during early winter at stations NPF, with a second peak in deeper waters, as observed by Jacquet et al. (2008b) during the iron (Fe) fertilization experiment (EIFEX). The deeper peak could also be linked to relatively larger cells that sink faster as they remineralise, possibly a large bloom early in the season.

Please revise your discussion and conclusion according to these comments.

Response: We have extensively revised the discussion and conclusion of the manuscript to better relay and support our study, and we sincerely hope that our revisions will be more convincing in this regard.

---

## Author Comment (AC3)

**Comment on bg-2021-42**

J.K.B. Bishop (Referee)

Referee comment on "Early winter barium excess in the Southern Indian Ocean as an annual remineralisation proxy (GEOTRACES GIPr07 cruise)" by Natasha René van Horsten et al., Biogeosciences Discuss., https://doi.org/10.5194/bg-2021-42-RC3, 2021

Review. van Horsten et al. "Early winter 1 barium excess in the Southern Indian Ocean as an annual remineralisation proxy".

The authors describe particulate Barium, O2, and potential density profile data from 7 stations in the Southern Ocean along from 59°S to 41°S crossing the Antarctic polar front (51°S) along 30°E south of Africa during GEOTRACES GIPr07 (in early wintertime conditions, June 28-July 13). This is a hard to get and interesting data set. The hypothesis is that since particulate barium should have only a short residence time (days to weeks) in the water column the inventory of particulate Ba would be far lower at times of low productivity that at other times of the year. The authors report particulate Ba concentrations as high as seen in other seasons and infer an active biological carbon pump year-round. The stocks are regressed against annual mean primary production. Comparisons are made with other data sets from the Southern Ocean.

What I like about the work is the heroic effort to achieve sampling in the wintertime and the excellent primary data arising from the expedition. Also, the goal of finding the correct transfer function relating the inventory of particulate barium in the mesopelagic (an indicator of export) to remotely sensed biomass or primary productivity would be a big plus.

That said, the paper falls short of its goals. The regressions in Figure 3, and manuscript discussion provide no insight. The data south of the polar front are aliased by cloud obscured retrievals of surface chlorophyll and primary productivity (See e.g., Ocean color monthly composites) and fall on a different slope than north of the PFZ. The STZ station is an outlier. The discussion does not sufficiently unify these divergent observations.

Response: We thank Pr. Jim Bishop for his review of our manuscript.

Please see below our responses to the revisions and comments (in blue) and the excerpts from the revised manuscript (in red).

We agree that remotely sensed PP does have inherent issues, as for any scientific methodology, that being said, we have confidence in our data as we made use of OC-CCI data which integrates all available sensors. This approach increases the chance of accurate measurements compared to a single sensor approach.

We checked the percentage of valid pixels for all stations of the compilation dataset in order to prove that remotely sensed data south of the polar front (82 ± 29 %; mean ± SD, n = 488) is as valid as north of the polar front (90 ± 20 %; mean ± SD, n = 370). This information has been added to the manuscript under section 2.5.

In order to assess the validity of the remotely sensed PP data and demonstrate no meridional bias across the SO, the percentage valid pixels were calculated for data north (90 ± 20 %; mean ± SD, n = 370) and south (82 ± 29 % mean ± SD, n = 488) of the PF.

Reflecting on other reviewer comments, I am convinced that a more comprehensive analysis of the data (now abundant) from multiple projects need to be considered. I am sure that everyone referenced would have data to share.

Response: In the initial manuscript, the $Ba_{xs}$ data were plotted against satellite-derived PP, integrated over months prior to sampling. At the time of previous cruises (e.g., INDIGO 3, ANT X/6 and EPOS 2, as suggested by the two other reviewers), there was no remotely sensed PP data available. The PP data, when available, were measured during the cruise and are not representative of integrated PP over months preceding the study. This is why these data were not initially included in our compilation.

Nevertheless, the mesopelagic $Ba_{xs}$ stock (µmol m$^{-2}$) is now plotted against day of year sampled (Figure 4, see below) in the revised manuscript, including the INDIGO 3, ANT X/6 and EPOS 2 data. However, as stated in the figure caption and the main text, these data must be considered with caution because these samples were not digested using HF. This can lead to an underestimation of aluminium concentrations and an overestimation of $Ba_{xs}$, where there are possible significant lithogenic inputs (e.g., close to Antarctica and downstream of the Drake Passage).

[Figure]

**Figure 4: (a) Positions of $Ba_{xs}$ observations compiled from all known SO studies, on a cylindrical equal-area projection of the SO, the three SO basin cut offs are indicated by the dashed black lines, from left to right, Pacific, Atlantic and Indian. Integrated mesopelagic $Ba_{xs}$ stock plotted against day of year sampled, with the 1st of September set as day 1, for all available literature data and winter data from this study. Data was split into two zones using the Polar Front (PF) to divide the SO; (b) South of the PF (SPF) and (c) North of the PF (NPF). Monthly averaged remotely sensed PP plotted against day of year, for locations and dates of the SO compilation dataset and winter data from this study; (d) SPF and (e) NPF. Open circles are data points from studies which did not use HF in the particulate sample digestion procedure, regressions did not include these data.**

I echo a need for a fuller hydrographic and dissolved phase framework for data interpretation – the supplemental data are very sparse.

Response: The hydrographic data was originally available on an ftp site (address given on page 19, line 30). Unfortunately, we have been made aware of problems with the ftp site and we apologize for that.

Temperature, salinity and nutrient data have now been included in the supplementary as Table S5. Unfortunately, no dBa data are available (nor were samples collected for this parameter).

There are some issues: (1) methodology: bottle sampling and the missing large particle fraction,

Response: We are aware of the differences in sampling systems. Indeed, sampling spigots on Go-Flo bottles are located 4 cm above the bottom of the bottle, which precludes the extraction of water below that level, potentially leaving behind particles that settle below the point of extraction. Strategies to avoid, minimize, or account for these biases have been discussed (e.g., Planquette & Sherrell, 2012) and are recommended in the GEOTRACES cookbook protocols, which we strictly followed.

Moreover, all the data we are comparing have been generated from samples collected with bottles, not in-situ pumps making the comparison less biased in this aspect.

This is now included under section 2.3 of the revised manuscript.

Volumes of 2 to 7 L of seawater were filtered from the GO-FLO bottles onto acid-washed polyethersulfone filters (25 mm diameter, Supor, 0.45 μm pore size), mounted on swinnex filter holders, for pBa and pAl analyses. Filters were mounted in line on the side spigot of each Go-Flo bottle. Furthermore, bottles were mixed 3 times before filtration, as recommended by the Geotraces protocols. Although the large fast sinking fraction of particles may be under sampled by using bottles (Bishop and Edmond, 1976; Planquette and Sherrell, 2012), comparing data that were generated using the same, internationally validated sampling systems and protocols (Cutter et al. 2017) as we do in this study, minimizes the potential biases.

and (2) the hypothesis of expected low wintertime concentrations is premised on particle sinking rates that are far too high (50 m $d^{-1}$) for the micron sized particles that comprise the bulk of suspended barite (sinking speed $\sim 0.1$ m $d^{-1}$).

Response: The hypothesis of expected low wintertime concentrations was taken from literature (Jacquet et al., 2008b, 2011).

The sinking speed of 50 m $d^{-1}$ was referring to the sinking of aggregates from the surface layer, and not to the sinking speed of suspended barite. This value is within the range of sinking speeds reported in the literature for sub millimetre sized aggregates (50 - 100 m $d^{-1}$: Riebesell et al., 1991; 50 - 430 m $d^{-1}$ around South Georgia: Cavan et al. 2015; mean of $\sim$ 100 m $d^{-1}$ in the Southern Ocean as reviewed in Laurenceau-Cornec et al., 2015; 10 - 150 m $d^{-1}$: McDonnell and Buesseler, 2010).

The sinking rates were used to estimate the maximum time needed by sinking aggregates to exit the mesopelagic zone, i.e., 1000m depth:

time = 1000 m / 50 m $d^{-1}$ = 20 days

With a mean current speed of 0.2 m $s^{-1}$ (Ferrari and Nikurashin, 2010), aggregates sinking through the mesopelagic zone (above 1000 m depth) in 20 days would have

been displaced a maximum of 346 km eastward, towards the location of stations sampled for mesopelagic $Ba_{xs}$:

$$0.2 \text{ m s}^{-1} \times 3600 \times 24 = 17280 \text{ m d}^{-1}$$

$$17280 \text{ m d}^{-1} \times 20 \text{ days} = 345\ 600 \text{ m} = 346 \text{ km eastward}$$

This distance was used to define the area of the sampling window where we estimated surface PP that was likely to have formed the aggregates that were further responsible for the release of the mesopelagic $Ba_{xs}$ signal measured within the mesopelagic zone.

The low sinking speed of suspended barite (~0.3 m d$^{-1}$, Sternberg et al. 2008), once produced in the mesopelagic layer, implies that the residence time of barite is indeed long, as suggested by Pr. Bishop in the last comment of the review. To sink from 300 m depth (~peak of production) to the bottom of the mesopelagic layer (1000 m depth), it would take (1000-300) / 0.3 = 2300 days, i.e., ~ 6 years. This supports our hypothesis, that the $Ba_{xs}$ signal in the mesopelagic layer may represent remineralisation activity over more than a few days to weeks, contrary to what was previously reported in SO studies (Dehairs et al., 1997; Cardinal et al., 2005; Jacquet et al., 2007, 2008a). It also further explains how mesopelagic $Ba_{xs}$ concentrations increase from spring to a maximum in winter (see new Figure 4b and c).

These explanations are now included in the text, and we thank Pr. J. Bishop for this comment that assisted us in clarifying this part.

[revised manuscript text omitted]

A couple of issues further complicating review is simply the lack of any access to the more complete data sets from the cruise beyond those used in figure 2 or the partially complete data sets used in figure 3. The cruise data should be available as supplemental data and also submitted to the GEOTRACES archives and DOI traceable.

Response: Data was made available on the SOCCO ftp site (see page 19, line 30 of the preprint) but as stated before, we were made aware of issues with the ftp, and we apologize for this inconvenience. We have now included these data in two tables in the supplementary material, Table S3 for this study and literature data, and Table S4 for nutrients, temperature, and salinity data for this study. Our data will be submitted to the GEOTRACES IDP once published.

I think the fundamental logic flaw (see comments below) lies on page 12 in the discussion of inferred barite residence times in the mesopelagic. I don't see a major advance beyond referenced work and don't support publication of this paper with its present interpretive framework. I encourage the Authors to look again at the results in a larger framework.

Response: Although we agree that further interpretation of data was required to improve the manuscript and support the study, we also believe that this work does go beyond what has been previously published. This is to our knowledge the first $Ba_{xs}$ dataset obtained during winter in the SO and we also use all available SO data and link it to remotely sensed PP, which has not yet been done.

We have included a more extensive assessment of the data in the revised manuscript that we believe to be more convincing.

Figure 4 is not needed. It is out of place. There is room for more figures.... Some detailed comments follow.

Response: As the three reviewers agree on this point, we have decided to remove this figure from the revised manuscript.

P4 line 08, R2=0.83... This seems like a bad validation of the O2 results.

Response: The slope (0.94 ± 0.10), intercept (0.07 ± 0.51) and p-value (7 x 10$^{-10}$), of the calibration regression indicate that although the correlation coefficient is not as high as could be expected ($R^2$ = 0.83), the calibration is still valid. The sensor had also been calibrated by the manufacturer less than a year prior to the cruise (August 2016).

With this in mind, we do not use any absolute values of dissolved $O_2$, instead, the shape of the $O_2$ profiles were used to support evidence of mesopelagic remineralisation and oxygen consumption, and how it is linked to the $Ba_{xs}$ profiles.

This is now clearly stated in the manuscript.

Temperature (°C), salinity and dissolved $O_2$ (µmol L$^{-1}$) profiles were measured by sensors (SBE 911plus) which were calibrated by the manufacturer within a year prior to the cruise. At each cast discrete seawater samples were collected and analysed onboard for calibrating sensor salinity (8410A Portasal salinometer, $R^2$ = 0.99) and dissolved $O_2$ measurements (Metrohm 848 titrino plus; Ehrhardt et al., 1983, $R^2$ = 0.83).

Decreases in dissolved $O_2$ concentrations at intermediate depths, together with $Ba_{xs}$ concentrations, were used to define the mesopelagic remineralisation depth range.

p 4 line 10. The Python language did not exist in 1982. Please fix sentence.

Response: This sentence has been corrected, with the publication referring to the calculation.

Temperature and salinity measurements were used to calculate potential density ($\sigma_\theta$; Gill, 1982) to characterise water masses sampled and to identify the mixed layer depth (MLD).

P 5. Lines 18-20. I assume this was an in-line filter, directly connected to the side spigot of the bottle. State what was done. Also, state whether or not the large sinking particle fraction would be sampled.

Response: Indeed, there was an in-line filter (Supor) directly connected to the side spigot of the bottle. Bottles were mixed prior to filtering with the in-line filter as recommended by the Geotraces cookbook. Strategies to avoid, minimize, or account for these biases have been discussed (e.g., Planquette & Sherrell, 2012) and are recommended in the GEOTRACES cookbook. We strictly followed these protocols. This is now clearly stated in section 2.3.

Moreover, all the data we are comparing is generated from samples collected with bottles, not in-situ pumps making the comparison possible with studies included in the compilation dataset. This is also now clearly stated.

Filters were mounted in line on the side spigot of each Go-Flo bottle. Furthermore, bottles were mixed 3 times before filtration, as recommended by the Geotraces protocols.

Although the large fast sinking fraction of particles may be under sampled by using bottles (Bishop and Edmond, 1976; Planquette and Sherrell, 2012), comparing data that were generated using the same, internationally validated sampling systems and protocols (Cutter et al. 2017) as we do in this study, minimizes the potential biases.

p 5. line 29. If varying volumes of water were filtered, the blank will not be a constant value. (Ba(filter)-blank(filter))/volume filtered. to get Ba and error should be the

s.d/filter blanks / volume filtered. Or is this what you did? I think the calculation was done correctly as error bars vary in size. Please clarify methods.

Response: Yes, this is indeed what was done. Unused filters were used to subtract the blank contribution. This is now explained in section 2.3.

Unused blank filters and filters containing the samples were acid reflux digested at 130°C in acid-cleaned savillex vials using a mixture of HF and $HNO_3$ (both Ultrapure grade, Merck) solutions (Planquette and Sherrell, 2012).

Mean amounts (in nmol) of a given element determined in unused filter blanks were subtracted from the amounts in the sample filter then divided by the volume filtered.

ALSO please state the assumed particle size fraction that has been sampled. There is no evidence that bottles adequately sample the sinking particle fraction.

Response: As stated above, we are aware that bottles may under sample the large fast sinking fraction of particles. There are unfortunately, also other potential sources of discrepancy such as filtration pressure (Gardner et al., 2003; Liu et al., 2005), filter type (Bishop et al., 2012), breakage or leakage of phytoplankton and other cells (e.g., Collos et al., 2014), creation of particles (Liu et al., 2005). Unfortunately, no sampling method is assumed to be perfect. Comparing data that were generated using the same, internationally validated sampling systems and protocols, as we do, minimizes the potential biases

P 6 Line 63. "The data..." Which data?

Response: The integrated remotely sensed PP data. This is now clearly stated in the revised manuscript.

The integrated remotely sensed PP data were regridded to 0.25° spatially, using bilinear interpolation, and averaged monthly.

P9. Fig. 2: O2 scale too compressed to be useful. Authors should provide complete data as supplemental (not just pAl, pBa, Baxs) and submit as soon as possible to GEOTRACES. Include T, S, sigma theta, o2, nutrients, dissolved Ba...

Response: Figure 2 has been amended to widen the dissolved $O_2$ scale. The temperature, salinity, and nutrient data has now been included in the supplementary (Table S5). We intend on submitting the complete dataset to GEOTRACES IDP once published.

[Figure]

p10... .Lines 31-33. "When taking into account....". There is something wrong with this sentence.

Response: This sentence has been rephrased

A noticeable difference between profiles sampled early in the bloom season (Dehairs et al., 1997; Jacquet et al., 2015) versus those sampled later (Cardinal et al., 2001; Planchon et al., 2013), are the contrasted concentrations of $Ba_{xs}$ in the surface mixed layer.

p 10. Line 34. Very high values can be associated with Rhizosolienia blooms (Bishop, 1988).

Response: We agree that very high values can be associated with Rhizosolienia blooms, but also in most cases when productivity is high, specifically in the SO.

P 12. Lines 84 & 85. Line 94-95... "Residence time of barite in mesopelagic days to weeks. & Particle sinking speeds of 50 m d$^{-1}$". The large particles comprising the flux do sink that fast; however, the subsurface barite is produced by fragmentation of these particles as they sink. The resulting micron sized barites sink at 0.1 m d-1. Thus, the

premise of decay to background on the time scale of days to weeks is invalid. Barites in the mesopelagic would have a residence time (by sinking) of hundreds of days – if not years. The sink for these barites is dissolution and reaggregation. As grazing is reduced in the wintertime then dissolution and sinking would dominate.

Response: As explained at the beginning of the review, the sinking speed of 50 m d$^{-1}$ was indeed referring to the sinking of aggregates from the surface layer, and not to the sinking speed of barite crystals.

As stated above, the low sinking speed of suspended barite (~0.3 m d$^{-1}$, Sternberg et al. 2008), once produced in the mesopelagic layer, indeed implies that the residence time of barite is longer than a few days to weeks as generally postulated in the literature for the SO (Dehairs et al., 1997; Cardinal et al., 2005; Jacquet et al., 2007, 2008a). To sink from 300 m depth (~peak of production) to the bottom of the mesopelagic layer (1000 m depth), it would indeed take (1000-300) / 0.3 = 2300 days, i.e., ~ 6 years, without considering time needed for reaggregation and redissolution. This supports our hypothesis that the Ba$_{xs}$ signal in the mesopelagic layer represents remineralisation activity over more than a few days to weeks, as previously reported, and is not directly linked to synoptic measurements of PP during the same cruise. It also further explains how mesopelagic Ba$_{xs}$ concentrations increase from spring to a maximum in winter (see new Figure 4b & c) and suggests that the background value has to be measured at the transition between winter and spring, just before biological activity resumes.

These explanations are included in the text of the revised manuscript.

I've not addressed the detailed discussion further as this point and invalidates the key conclusion of the authors.

Response: We hope that our detailed responses and extensive revisions will convince Pr. Bishop of the validity of our hypothesis and manuscript.

jim Bishop (UC Berkeley).

p.s. have a look at Bishop 1989 (attached - since it may be hard to find). The mapped representation of barite stocks is virtually the same as sampled here.

Please also note the supplement to this comment:

https://bg.copernicus.org/preprints/bg-2021-42/bg-2021-42-RC3-supplement.pdf

---

## Author Response (AR2)

Referee #1: Frank Dehairs, fdehairs@vub.be

The authors have done a considerable effort addressing the comments of the reviewers and the manuscript has been considerably strengthened, especially regarding the discussion about the time scale of the Baxs signal.

Response: We thank Prof. Frank Dehairs for this very positive comment

The calculation of mesopelagic Baxs accumulation rates is an interesting outcome, but which is not really exploited further. Could authors say more about this, maybe in terms of the oceanic Ba cycle (ins, outs)?

Response: The $Ba_{xs}$ accumulation rate indeed reflects the balance between "ins" (remineralisation of organic material and advection) and "outs" (dissolution, advection and settling). If we assume minimal surface export remineralisation in the mesopelagic zone, during winter, we can have a rough approximation of the loss rate between winter and early spring (Figure 4b & c: going back to the "baseline" ~ 180 pmol $L^{-1}$). An extrapolation from measurements conducted during winter down to early spring observations results in an estimated loss rate of ~ 1 µmol $m^{-2}$ $d^{-1}$ and ~ 2 µmol $m^{-2}$ $d^{-1}$, south and north of the PF, respectively (see Figure below). That would give an estimated net $Ba_{xs}$ accumulation rate (i.e. gross accumulation rate + loss rate) of ~ 1.9 µmol $m^{-2}$ $d^{-1}$ and ~ 2.9 µmol $m^{-2}$ $d^{-1}$ during the productive season, south and north of the PF, respectively. However, these estimates remain very crude and we believe that these preliminary results should be further investigated, e.g. in a physico-biogeochemical model (which is beyond the scope of our paper). The figure showing the decline of mesopelagic $Ba_{xs}$ stock from winter down to early spring can be included as a supplementary figure if Prof. Dehairs wishes so, however, we do not feel that the regression is robust enough with the current available data spanning this timeframe.

[Figure]

In the supplementary material the heading of Table S4 has the sentence: "Where no POCrem fluxes are reported negative values were estimated ..." I guess this is an error? Please correct.

Response: We were initially not aware of the publication by Dehairs and Goeyens (1996). We thus calculated the mesopelagic POC remineralisation fluxes for the INDIGO 3 and EPOS 2 datasets (which were kindly provided by Prof. Dehairs) using the depth range (100 - 1000 m depth) and $Ba_{residual}$ value (180 pmol $L^{-1}$) as was used for our data and data from previous publications, where mesopelagic POC remineralisation fluxes were not published. This resulted in negative values for some stations where pBa concentrations was very low, due to the high $Ba_{residual}$ value used. Since we made use of published values for the compilation dataset, where available, we have now included the values as calculated by Dehairs and Goeyens (1996), where POC remineralisation fluxes were calculated using a 200 - 400 m depth range and 50 pmol $L^{-1}$ as the $Ba_{residual}$ value. This recalculation does not affect the results of our study, as these data were not included in the regressions due to the samples not being digested with HF and the observations being conducted prior to the availability of remotely sensed PP (Figure 4).

Referee #3: J.K.B. Bishop, jkbishop@berkeley.edu

Review. van Horsten et al. "Early winter barium excess in the Southern Indian Ocean as an annual remineralisation proxy".

The authors have made significant improvements to their manuscript and have addressed virtually all comments from reviewers.

Response: We thank Prof. Jim Bishop for this positive comment.

Here are 4 (Qx) important questions that I'd like the authors to address.

The concentration of micron sized barite particles in the water column reflects the balance between the rate of addition from fragmenting large aggregate particles and the rate of their loss due to combined effects of three processes: single particle sinking, dissolution, and finally, reaggregation and sinking due to actions of filter feeders living in the mesopelagic.

Question 1: If source and sink processes are variable with depth and season, and food web, why would the authors expect a constant "background" Baxs?

Response: The hypothesis of a "background signal" arises from measurements showing values close to 180 pmol L$^{-1}$ in deep water masses, previously reported multiple times in the Southern Ocean (Dehairs and Goeyens, 1996; Dehairs et al., 1997; Jacquet et al, 2008a; 2008b; 2011; 2015; Planchon et al., 2013).

That being said, the consistency of this value is still up for debate. It can be hypothesized that background Ba$_{xs}$ is mainly constituted of barite crystals. Indeed, considering the residence time of barite crystals of ~ 6 yrs in the mesopelagic layer (as estimated in the previous review), and considering a mean current speed varying from ~20 cm.s$^{-1}$ (500 m depth) to 14 cm.s$^{-1}$ (1975 m depth) in the ACC region (Vigo et al, 3D Geostrophy and Volume Transport in the Southern Ocean, Remote sensing, 2018), barite crystals can be transported over 26 000 to 38 000 km before settling out of the mesopelagic zone. With the circumference of the Southern Ocean being ~ 20 000 km, these small particles may have enough time to be mixed throughout the deep waters of the Southern Ocean, homogenizing the background signal that would not reflect the variability in surface PP, food web, etc.

A better characterization of the Ba particles along the water column, and physical speciation, together with more accurate estimates of the sinking speed of these particles, would be necessary to confirm this hypothesis. That being said, measurements so far do indicate a "background" value for Ba$_{xs}$ that is close to 180 pmol L$^{-1}$ for studies conducted throughout the year (Dehairs and Goeyens, 1996; Dehairs et al., 1997; Jacquet et al, 2008a; 2008b; 2011; 2015; Planchon et al., 2013).

In Dehairs et al. (1997; DH1997) the regression of Baxs(averaged 200-400 m) vs estimated O2 consumption rate yields the equation: Baxs = 218 * 10(20.01*O2consumption), where 218 pM Ba is the intercept, or "Background" value. DH1997 justifies the O2 vs Ba relationship with data shown in their Fig 5 where there is a similarity of profiles of O2 consumption rate and Baxs.

In this paper, The authors have recast the same 7 points used in DH1997 to a linear relationship: $JO2 = (Mesopelagic\ Ba\text{xs} - Ba\text{residual})/17200$ in equation (3); Where

*Mesopelagic Ba*xs is depth-weighted Baxs from the base of the euphotic layer to 1000m. Furthermore, they have assigned Babackground = 180 pM. In this paper, Equation (2) scales Jo2 by the thickness of the mesopelagic zone (~900m), the Redfield C:O2 ratio and by 12.01, the atomic weight of carbon to yield mesopelagic POC remineralization.

Question 2: How do the authors justify applying the DH1997 transfer function with results calculated in a fundamentally different fashion?

Response: Through personal communication with Prof. Dehairs, who has also reviewed this manuscript, it was confirmed that there were two transfer functions, one linear (Dehairs and Goeyens, 1996) and one exponential (Dehairs et al., 1997). It was confirmed that both functions gave very similar results, and it was therefore decided to use the simpler, linear version (Dehairs, per. comm.), which is what has been used in all publications since, with POC remineralisation fluxes corresponding well to estimates using the $^{234}$Th method in the Southern Ocean and the North Atlantic (Lemaitre et al., 2018; Planchon et al., 2013). The linear function was reassessed in the North Atlantic by Lemaitre et al. (Figure 8: 2018, below), as well as more recently in the Mediterranean Sea by Jacquet et al. (2021).

[Figure]

**Figure 8.** Regression of DWA mesopelagic Ba$_{xs}$ (pmol L$^{-1}$) versus O$_2$ consumption rate (µmol L$^{-1}$ d$^{-1}$) using the Southern Ocean transfer function from Dehairs et al. (1997; red circles) and the transfer function obtained here for the North Atlantic (black circles). Station 44 (triangle) was excluded from the regression. If station 44 is included, $R^2 = 0.33$ and the $p$ value $= 0.07$.

We have amended the citation in the manuscript to refer to Dehairs and Goeyens (1996) as well as the Dehairs et al. (1997) publication. All data used in our manuscript made use of the linear function. The depth range over which the Ba$_{xs}$ is integrated varies between studies due to where the Ba$_{xs}$ peak is detected in the water column. We integrated Ba$_{xs}$ over varying depth ranges from the base of the mixed layer covering the specific peak depth range, down to 1000m, 1500m and down to deep waters, with no significant differences between the mesopelagic POC remineralisation fluxes obtained, using the different depth ranges. We therefore decided to use the operationally defined mesopelagic depth range (100 - 1000m: Robinson et al., 2010) for our calculations.

As previously stated, we made use of the standard $Ba_{residual}$ concentration used in previous Southern Ocean studies included in the compilation dataset (Jacquet et al, 2008a; 2008b; 2011; 2015; Planchon et al., 2013).

Indeed, later calculations by Jacquet et al. (2008; DSR) refined the background value using the saturation state of the water with respect to barite (Monnin et al., 1999, Monnin and Cividini, 2006). This has been found to be in agreement with other Southern Ocean studies (Jacquet et al, 2008a; 2008b; 2011; 2015; Planchon et al., 2013), including our study (see Results : "when averaging all concentrations below 2000 m along the transect, the $Ba_{residual}$ concentration was $161 \pm 43$ pmol $L^{-1}$ (mean $\pm$ SD, n = 15)." )

P. 10 lines 255 – 260: The mean $Ba_{residual}$ concentration south of PF was $183 \pm 29$ pmol $L^{-1}$ (mean $\pm$ SD, n = 7), whereas it was $142 \pm 45$ pmol $L^{-1}$ (mean $\pm$ SD, n = 8) between the PF and the STF. The two regions were however not significantly different to each other when conducting a Welch's t-test (t-statistic = 2.10; p-value = 0.06) and when averaging all concentrations below 2000 m along the transect, the $Ba_{residual}$ concentration was $161 \pm 43$ pmol $L^{-1}$ (mean $\pm$ SD, n = 15). This concentration is not statistically different from the literature value of 180 pmol $L^{-1}$ (Jacquet et al, 2008a; 2008b; 2011; 2015; Planchon et al., 2013), which is widely used for estimates of POC remineralisation fluxes.

P. 12 lines 311 – 314: In deeper waters along the transect, south of the STF, (below 2000 m) where remineralisation is minimal compared to the mesopelagic zone, our $Ba_{xs}$ concentration of $161 \pm 43$ pmol $L^{-1}$ (mean $\pm$ SD, n = 15) is not significantly different from the widely used $Ba_{residual}$ concentration of 180 pmol $L^{-1}$, measured during early Spring to late Summer (e.g., Jacquet et al., 2008a; 2008b; 2011; 2015; Planchon et al., 2013).

Question 3: Can the authors demonstrate that their data or other datasets fall on the same trend as in Fig. 5 in DH1997? It is fundamentally important to the paper to make this important logical transition.

Response: The correction has been made to the citation, to refer to the linear transfer function published by Dehairs and Goeyens (1996). The linear function has been used in all Southern Ocean publications since. We are currently not capable of replicating this function as we do not have access to the required oxygen utilization rate data. It was, however, done by Lemaitre et al. (2018), in the North Atlantic, and no significant difference was found for their data.

Question 4: Why would the authors expect the DH1997 transfer function to apply across the entire southern ocean domain?

Response: The linear transfer function was obtained using data from across various environments, characterized by different regimes of plankton community composition and productivity (Dehairs and Goeyens, 1996).

The linear transfer function has also been used successfully and validated across biogeochemical zones and basins of the Southern Ocean (Cardinal et al., 2005; Planchon et al., 2013), as well as in the North Atlantic (Lemaitre et al., 2018) and recently in the Mediterranean Sea (Jacquet et al., 2021).

In summary, I thank the authors for their efforts so far. The expanded data analysis/inclusion of other data sets greatly improves the paper. That said, please address the above questions prior to publication.